# A versatile method for computing optimized snow albedo from spectrally fixed radiative variables : VALHALLA v1.0

Florent Veillon [1,2], Marie Dumont [2], Charles Amory [1,3], and Mathieu Fructus [2]

[1]Laboratory of Climatology, Department of Geography, SPHERES, University of Liège, Liège, Belgium
[2]Université Grenoble Alpes, Université de Toulouse, Météo-France, CNRS, CNRM, Centre d'Etudes de la Neige, 38000 Grenoble, France
[3]Université Grenoble Alpes, CNRS, Institut des Géosciences de l'Environnement, 38000, Grenoble, France

**Correspondence:** Marie Dumont (marie.dumont@meteo.fr)

**Abstract.** In climate models, the snow albedo scheme generally calculates only a narrowband or broadband albedo, which leads to significant uncertainties. Here, we present the Versatile ALbedo calculation metHod based on spectrALLy fixed radiative vAriables (VALHALLA, version 1.0), to optimize spectral snow albedo calculation. For this optimization, the energy absorbed by the snowpack is calculated by the spectral albedo model Two-streAm Radiative TransfEr in Snow (TARTES) and the spectral irradiance model Santa Barbara DISORT Atmospheric Radiative Transfer (SBDART). This calculation takes into account the spectral characteristics of the incident radiation and the optical properties of the snow, based on an analytical approximation of the radiative transfer of snow. For this method, 30 wavelengths, called tie points ($tps$), and 16 reference irradiance profiles are calculated to incorporate the absorbed energy and the reference irradiance. The absorbed energy is then interpolated for each wavelength between two $tps$ with adequate kernel functions derived from radiative transfer theory for snow and the atmosphere. We show that the accuracy of the absorbed energy calculation primarily depends on the adaptation of the irradiance of the reference profile to that of the simulation (absolute difference < 1 $\mathrm{W\,m^{-2}}$ for broadband absorbed energy and absolute difference < 0.005 for broadband albedo). In addition to the performance in accuracy and calculation time, the method is adaptable to any atmospheric input (broadband, narrowband), and is easily adaptable for integration into a radiative scheme of a global or regional climate model.

## 1 Introduction

The solar irradiance is an essential source of energy to snow and ice surfaces (Warren, 1982). Absorption of shortwave radiation strongly depends upon the physical properties of snow and atmospheric conditions. The albedo, defined as the fraction of reflected solar radiation, is very high for fresh snow and limits energy absorption by the snowpack. Darker or old snow and glacial ice absorb more solar energy (Warren, 1982; Gardner and Sharp, 2010). The snowpack may also contain light absorbing particles (LAPs, McKenzie Skiles and Painter, 2018), leading to a decrease in albedo (Warren, 1982; Picard et al., 2009; Gardner and Sharp, 2010; Libois et al., 2013; Dumont et al., 2014). Besides, the optical properties of snow and ice strongly vary with the wavelength (e.g. ice refraction index of Warren and Brandt, 2008). The snow spectral albedo, defined as the fraction between reflected and incident solar energy for a given wavelength (Grenfell et al., 1994), is higher for near-ultraviolet

(near-UV, 300–400 nm), visible (400-700 nm) and near-infrared (near-IR, 750–1400 nm) but is lower in the IR part of the solar
spectrum (Warren, 1982; Gardner and Sharp, 2010). The changes in albedo with snow and ice properties play a major role
in the melt-albedo feedback (Cess et al., 1991). An increase in temperature favours rapid metamorphism and melting of the
snow cover, which leads to coarser snow grains and less reflective surface. More energy is then absorbed and made available
for heating the snowpack, enhancing snow metamorphism and melting (e.g. Flanner and Zender, 2006). The solar zenith angle
(SZA, valid for direct radiation) and atmospheric conditions (e.g. clouds, aerosols loads, water vapour column), determine
the amount of energy reaching the surface of the snowpack. For example, clouds influence the proportion of solar radiation
reaching the surface and contribute to total incident radiation by emitting longwave radiation (Wetherald and Manabe, 1988;
Schneider et al., 2019). For realistic estimates of the energy balance and melt over snow and ice surfaces, accurate knowledge
of a set of atmospheric and snowpack properties is thus required (Picard et al., 2012).

In addition to the above-mentioned requirements, accuracy in the estimation of the energy absorbed at the snow surface
can be achieved through spectral calculation of the albedo but remains numerically expensive. This also requires spectral
calculations of the solar irradiance that are most of the time not available in climate models. This is usually overcome in
most global and regional climate models by computing broadband or narrowband albedo to estimate the energy budget at the
snow and ice surfaces (Gardner and Sharp, 2010; Kuipers Munneke et al., 2011). The broadband albedo is defined as the ratio
between total reflected and total incident solar energy integrated across the entire solar spectrum, whereas the narrowband
albedo is integrated over a limited range of the solar spectrum. These integrations however lead to a bias in the calculation of
the snowpack albedo, which ultimately propagates in the computation of the surface energy and mass budgets.

To overcome these uncertainties while maintaining an adequate calculation time to remain competitive, new methods are
developed. One of them, recently developed by (Van Dalum et al., 2019), effectively couples a snow spectral albedo model
with a narrowband atmospheric radiation scheme. This method (Spectral-to-NarrOWBand ALbedo module; SNOWBAL) al-
lows the coupling of the radiative transfer model TARTES (Two-streAm Radiative TransfEr in Snow, Libois et al., 2013) with
the European Centre for Medium-Range Weather Forecasts (ECMWF) radiation McRad scheme based on the shortwave Rapid
Radiation Transfer Model (RRTMsw) embedded in RACMO2 (Mlawer et al., 1997; Clough et al., 2005; Morcrette et al.,
2008; ECMWF, 2009). They used the first 12 of the 14 predefined representative wavelengths (RWs, for every 14 bands of
RRTMsw) dependent on irradiance distribution and albedo within a spectral band to calculate the narrowband albedo and radi-
ation absorption in each (sub)surface snow layer. To determine the 12 RWs, a limited number of properties of the atmosphere
are selected using a look-up table (LUT). They demonstrate that RWs primarily depend on the SZA, cloud content and water
vapour. This method is tested on different types of snow and for clear-sky and cloudy atmospheric conditions, and represents
broadband snow albedo with low uncertainties (<0.01). In van Dalum et al. (2020a, b), the SNOWBAL module is evaluated
in RACMO2 over the Greenland ice sheet. This method can therefore be used on large surfaces while accurately representing
the albedo of snow and ice. The impact of snow properties on the RWs are not accounted for in the LUTs since it is negligible
in the case tested in Van Dalum et al. (2019). This might be a limitation when the narrow bands of the atmospheric model are
too large. The use of this method with another model than RACMO2 will require a recalculation of the LUTs for a different

set of narrowbands. The accuracy of the method is also expected to increase if more narrowbands are available, reducing the sub-band spectral variability.

Here, we describe a novel method for calculating accurately the solar energy absorbed by the snowpack based on the determination of spectrally fixed radiative variables. The method is named VALHALLA for Versatile ALbedo calculation metHod based on spectrALLy fixed radiative vAriables (version 1.0). This method maintains adequate accuracy of absorbed energy values while reducing calculation time irrespective of the radiative transfer scheme used for the atmosphere. While VAL-HALLA as SNOWBAL is a coupling scheme, VALHALLA fulfills a different niche than SNOWBAL since it allows accurate

calculation when only broadband atmospheric inputs are available and accounts for the snow properties variations. SNOWBAL requires accurate snow radiative transfer calculations for a limited number of wavelengths and an adequate representation of the atmosphere, i.e., cloud content, water vapour, sza, direct-to-diffuse irradiance ratio. VALHALLA requires accurate radiative transfer calculations for both snow and atmosphere for a limited number of wavelengths. The proposed method takes advantage of the spectral characteristics of incident radiation and optical snow properties, based on the analytical approximation of the

radiative transfer within the snowpack provided by Kokhanovsky and Zege (2004). The accuracy of the methods is assessed using accurate calculation at a spectral resolution of 1 nm. The sensitivity of the albedo calculations to the atmospheric and snow properties is also assessed. The results are compared with reference albedo calculations at different spectral resolutions and with other existing methodologies (Van Dalum et al., 2019; Gardner and Sharp, 2010). Implementation considerations in climate and land models are finally discussed.

## 2    Method

The VALHALLA method relies on the accurate calculations of the solar radiation absorbed by the snowpack for a small number of selected wavelengths, named tie points ($tps$) in the following. The number of tie points is kept as small as possible to limit the computing resources. Between these tie points, the VALHALLA method interpolates the absorbed radiation based on kernel functions that reflect the main spectral variations of the absorbed radiation across the solar spectrum. The general reasoning of

the method consists in assuming that the spectral variation between tie points can be approximated using the refractive index of ice. The calculation of the absorbed radiation at the tie points can be performed with any radiative transfer model. In the following, we selected the Two-streAm Radiative TransfEr in Snow (TARTES, Libois et al., 2013) for the snowpack and the Santa Barbara DISORT Atmospheric Radiative Transfer (SBDART, Ricchiazzi et al., 1998a), for the atmosphere.

### 2.1    Radiative transfer models

### 2.1.1    Radiative transfer in snow, TARTES

TARTES calculates spectral albedo in a multilayer snowpack when the physical properties of each layer and the angular and spectral characteristics of the radiation are known (Libois et al., 2014) and is embedded in the detailed snowpack model Crocus (e.g. Tuzet et al., 2017). TARTES is based on the Kokhanovsky and Zege (2004) formalism for weakly absorbing media to

describe the single-scattering properties of each layer and the delta-Eddington (Joseph et al., 1976) approximation to solve the radiative transfer equation. TARTES represents the snowpack as a stack of horizontal homogeneous layers. For calculation, physical properties of the snowpack (e.g., grain radius, grain shape, density, thickness, type of LAPs, LAPs concentration) and SZA for direct radiation (for diffuse radiation, SZA is fixed at 53°) are used as inputs. The grain size is characterized by the snow specific surface area (SSA; expressed in $m^2\,kg^{-1}$, defined as the ratio between the surface of the air-ice interface $S$ and the ice mass volume $V$) :

$$\text{SSA} = \frac{S}{V\rho_{\text{ice}}} \tag{1}$$

with $\rho_{\text{ice}}$, the volumetric mass of ice ($917\,\text{kg m}^{-3}$).

We used two shape parameters that are relevant for the optical properties of snow : the asymmetry parameter $g$ (dimensionless) and the absorption enhancement parameter $B$ (dimensionless, Libois et al., 2013). $g$ quantifies the amount of radiation that is scattered forward for a snow grain and $B$ quantifies the lengthening of photon paths inside a snow grain due to internal multiple reflections.

In Tuzet et al. (2017) and later studies, TARTES was used for calculations of radiative transfer with a spectral resolution of 20 nm. This resolution is the best compromise between the accuracy of radiation and calculation time, which is still very important, and makes this model configuration computationally expensive.

### 2.1.2 Radiative transfer in the atmosphere, SBDART

The model Santa Barbara DISORT Atmospheric Radiative Transfer (SBDART, Ricchiazzi et al., 1998b) is used for radiative transfer calculation in clear-sky and cloudy conditions in the atmosphere. SBDART uses Discrete Ordinate Radiative Transfer (DISORT, Stamnes et al., 1988) to solve the radiative transfer equation in the atmosphere vertically homogeneous. This model is organized to permit up to 65 atmospheric layers and 40 radiation streams. The main input parameters used in this study are the aerosol optical depth (AOD), the cloud-layer optical depth ($\tau$), the boundary layer aerosol type selector (IAER) and SZA. With these parameters, SBDART calculated direct and diffuse irradiance ($W\,m^{-2}$) for each wavelength between 0.320 and 4.000 µm. This atmospheric radiative transfer model was chosen since it provides accurate simulations of solar irradiance in snow covered areas (e.g., Tuzet et al., 2020) and offers a large number of parameters to set for the atmospheric properties.

### 2.2 The VALHALLA method

### 2.2.1 Theoretical basis

The spectral direct albedo, also called directional-hemispherical reflectance, $r$ of a homogeneous, optically infinite snowpack can be approximated by the following relationship (Libois et al., 2013; Dumont et al., 2017; Kokhanovsky et al., 2018) :

$$r(\lambda) = \exp\left(-u(\mu_0)\sqrt{\frac{64\pi}{3\rho_{\text{ice}}\text{SSA}(1-g)}\left(\frac{2n(\lambda)B}{\lambda} + 3\frac{\rho_{\text{ice}}}{\rho_{\text{LAP}}}c_{\text{LAP}}C_{abs}^{\text{LAP}}(\lambda)\right)}\right), \tag{2}$$

where $u(\mu_0) = \frac{3}{7}(1 + 2\mu_0)$; $\mu_0$ is the cosine of the solar zenith angle, $n(\lambda)$ is the imaginary part of the ice refractive index at the wavelength $\lambda$, $c_{\mathrm{LAP}}$ is the light absorbing particles concentration, $\rho_{\mathrm{LAP}}$ is the volumetric mass of the light absorbing particles, and $C_{abs}^{\mathrm{LAP}}(\lambda)$ is the absorption cross-section of LAPs.

When neglecting the spectral variations due to the presence of LAPs in Eq. 2, the first order of spectral albedo variations can be approximated as

$$r(\lambda) \sim exp(-J\sqrt{\frac{n(\lambda)}{\lambda}}), \tag{3}$$

where $J = u(\mu_0)\sqrt{\frac{128\pi}{3\rho_{\mathrm{ice}}\mathrm{SSA}(1-g)}}B$. $J$ is constant with $\lambda$ and depends only on SZA and snow physical properties.

The fraction of absorbed energy in the snowpack with respect to the incoming energy, $f_p$, is thus related to the spectral albedo by the following relationship :

$$f_p(\lambda) = 1 - r(\lambda), \tag{4}$$

For the atmosphere, we use the Beer-Lambert law to express the first order spectral variations of the incoming solar radiation. The Beer-Lambert's law establishes a relationship between the radiation transmitted through a given medium $I$ and the incident irradiance $I_0$ at the wavelength $\lambda$. Let $L$ be the thickness of the media and $c_a$ the absorption coefficient. Then :

$$I(\lambda) = I_0(\lambda) \exp(-c_a(\lambda)L), \tag{5}$$

$c_a(\lambda)$ is varying with the atmospheric profile, i.e. with the aerosols properties, the properties of gas in the atmosphere (water vapour, ozone, ...), the solar zenith angle and the cloud properties. Here we assume that the spectral variations of the solar irradiance at the surface can be written:

$$I(\lambda) \sim E_{ref}(\lambda) \exp(-D\sqrt{\frac{n(\lambda)}{\lambda}}), \tag{6}$$

where $E_{ref}$ is the total incident energy at the wavelength $\lambda$ for a reference atmospheric profile and $D$ is constant with wavelength. In other words, this means that for instance, for a given location, we assume that changes in the atmospheric solar irradiance with time can be modelled using Eq. 6, i.e. main spectral changes are driven by the water vapour (assuming that the refractive index of water vapour is close to $n(\lambda)$).

As a consequence, we assume that the absorbed energy by a snowpack for a given wavelength $E_{abs}(\lambda)$ can be approximated by :

$$E_{abs}(\lambda) \sim E_{ref}(\lambda) \exp(-D\sqrt{\frac{n(\lambda)}{\lambda}}) f_p(\lambda). \tag{7}$$

### 2.2.2 Interpolation method

The VALHALLA method is based on precise calculation of the absorbed energy at the tie points ($tps$), and on the interpolation between these wavelengths based on the general shape of the spectrum given in the equation above (Eq. 7). The method

calculates the absorbed energy since this is the variable directly used in the energy budget of the snowpack. The snow albedo can be directly diagnosed from the absorbed energy (e.g. Eq. 4).

The method uses a reference irradiance profile with a spectral resolution of 1 nm, $E_{ref}$. For each SZA value (varying between 10 and 80 degrees), a reference irradiance profile is calculated with SBDART. In total, 16 reference profiles were used
150 (one set for clear-sky and partially cloudy conditions and the other one for full overcast conditions; see Section 2.5.2). These profiles are used to calculate a coefficient $C$ between the broadband reference irradiance $E_{ref}$ (integral of the reference profile) and narrow or broadband irradiance $E_{exp,i}$ given by an atmospheric model for each narrow or broad spectral band $i$:

$$C_{b_i}^{b_{i+1}} = \frac{E_{exp,i}}{\int_{b_i}^{b_{i+1}} E_{ref}(\lambda)d\lambda}, \tag{8}$$

where $b_i$ and $b_{i+1}$ are the max and min wavelengths of the bands in which the atmospheric model is providing the solar
155 incident radiation. $C_{b_i}^{b_{i+1}}$ thus represents the scaling factor between the incident radiation provided by the atmospheric model and the reference irradiance for each narrow or broad spectral band of the atmospheric model. In the following, we use $b_i =$ 320 nm and $b_{i+1}$ = 4000 nm. Thus we assume that the broadband incident radiation only is available from the atmospheric model.

For each $tp$, the absorbed energy and irradiance are calculated using TARTES-SBDART and used for determining the values
160 of variables $D$ and $J$.

Between two tie-points $tp_n$ and $tp_{n+1}$, we assume that the absorbed energy can be approximated by :

$$E_{abs}(\lambda_{tp_n}^{tp_{n+1}}) = C_{b_i}^{b_{i+1}} E_{ref}(\lambda_{tp_n}^{tp_{n+1}}) \, exp(-D_{tp_n}^{tp_{n+1}} \sqrt{\frac{n(\lambda_{tp_n}^{tp_{n+1}})}{\lambda_{tp_n}^{tp_{n+1}}}}) \, (1 - exp(-J_{tp_n}^{tp_{n+1}} \sqrt{\frac{n(\lambda_{tp_n}^{tp_{n+1}})}{\lambda_{tp_n}^{tp_{n+1}}}})). \tag{9}$$

To determine these variables, which take into account all snow and illumination properties, an optimization by the least-square method is used. Indeed, $D$ and $J$ are mutually dependent.
165 In the context of optimization, variable $D$ is written in :

$$G_{tp_n}^{tp_{n+1}} = exp(-D_{tp_n}^{tp_{n+1}} \sqrt{\frac{n(tp_n)}{tp_n}}), \tag{10}$$

with :

$$D_{tp_n}^{tp_{n+1}} = -log( G_{tp_n}^{tp_{n+1}} ) \sqrt{\frac{tp_n}{n(tp_n)}}, \tag{11}$$

and $J$ :

170 $$J_{tp_n}^{tp_{n+1}} = -log( 1 - \frac{E_{abs_{tp_n}}}{C_{b_i}^{b_{i+1}} E_{ref_{tp_n}} G_{tp_n}^{tp_{n+1}}} ) \sqrt{\frac{tp_n}{n(tp_n)}}. \tag{12}$$

The optimization is realised on the variable $G_{tp_n}^{tp_{n+1}}$ and uses absorbed energy $E_{abs}$ and total irradiance $E_{ref}$ for $tp_{n+1}$.

$$\Delta_{tp_n}^{tp_{n+1}} = E_{abs_{tp_{n+1}}} - C_{b_i}^{b_{i+1}} E_{ref_{tp_{n+1}}} G_{tp_n}^{tp_{n+1}} (1 - exp(-J_{tp_n}^{tp_{n+1}} \sqrt{\frac{n(tp_{n+1})}{tp_{n+1}}})). \tag{13}$$

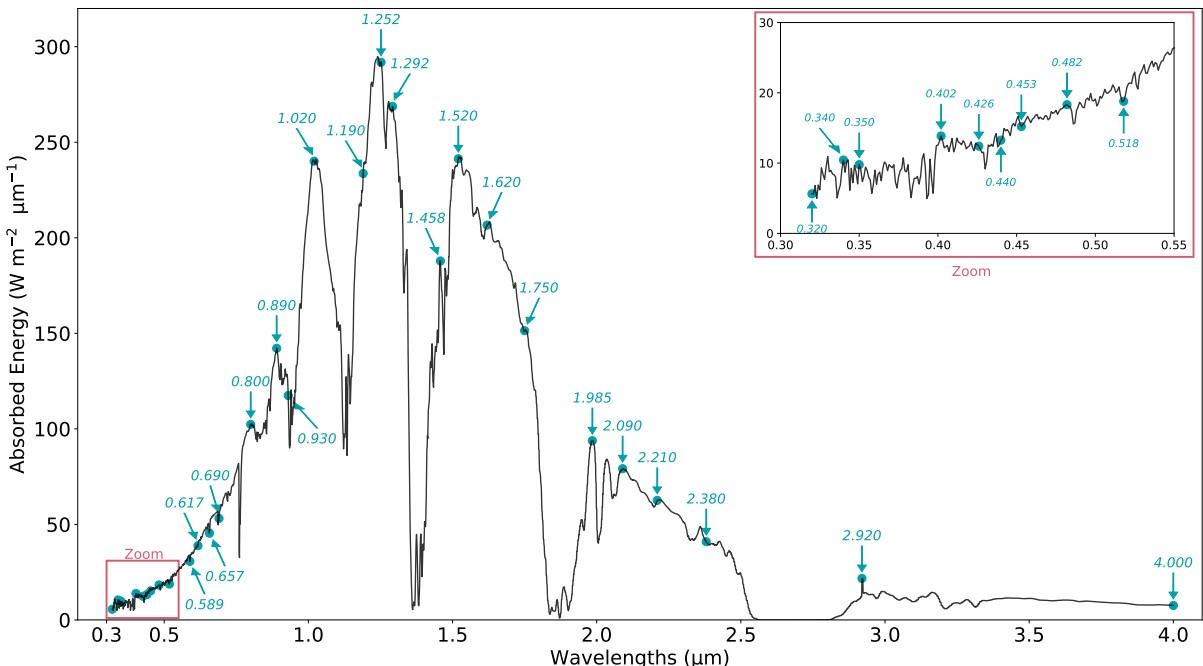

**Figure 1.** Spectral positions of $tps$ on an example of absorbed energy profile by a snowpack without LAPs.

Namely, an optimization method is used to solve Eq. 9. The optimization algorithm is finding the value of $G_{tp_n}^{tp_{n+1}}$ for which $\Delta_{tp_n}^{tp_{n+1}}$ is the closest to zero, $\Delta_{tp_n}^{tp_{n+1}}$ being the difference between the left and the right sides of Eq. 9.

### 2.2.3 Numericals settings

**Selection of the tie points**

The tie points, $tps$, are the reference wavelengths for absorbed energy and total irradiance. For all types of snow and cloud cover, a total of 30 $tps$ are selected as a compromise between accuracy and computational time (Fig. 1). The $tp$ has been selected as the local maxima and minima of absorbed energy after several optimization tests (not shown).

**Reference irradiance profiles**

To account for a representative set of atmospheric conditions, different reference irradiance profiles depending on SZA and cloud cover are chosen. These profiles are calculated by SBDART simulations with a spectral resolution of 1 nm for two cloud cover types. For simulations of clear-sky and partly cloudy conditions, reference irradiance profiles with values of $\tau$ equal to 0.5 are calculated. For simulations of full-overcast conditions, these profiles are calculated with a value of $\tau$ equal to 10 (Table 1).

**Table 1.** Atmospheric parameters of reference irradiance profiles. Each irradiance profile is calculated for eight values of SZA and two values of $\tau$. The other parameters are fixed in the SBDART model.

| Cloud cover conditions | Clear-sky and partially cloudy | Full overcast |
|---|---|---|
| Solar zenith angle (°, SZA) | 10 ; 20 ; 30 ; 40 ; 50 ; 60 ; 70 ; 80 | |
| Boundary layer aerosol type (IAER) | 2 - urban | |
| Aerosol optical depth (AOD) | 0.07 | |
| Cloud-layer optical depth ($\tau$) | 0.5 | 10 |
| Integrated ozone concentration(atm-cm) | 0.3 | |
| Integrated water vapor amount ($\mathrm{g\,cm^{-2}}$) | 0.35 | |
| Surface altitude (km) | 2.1 | |
| Optical depth of each stratospheric aerosol layer | 0.013 | |
| Atmospheric profile | 3 - Mid-Latitude Winter | |

**Table 2.** Atmospheric parameters of simulations. Each irradiance profile is calculated for eight values of the SZA, five values of IAER, three values of the AOD and five values of $\tau$ (one for clear-sky conditions, three for partially cloudy conditions and one for full-overcast conditions). The other parameters are fixed in the SBDART model.

| | | | | | |
|---|---|---|---|---|---|
| Solar zenith angle (°, SZA) | 10 ; 20 ; 30 ; 40 ; 50 ; 60 ; 70 ; 80 | | | | |
| Boundary layer aerosol type (IAER) | 0 no boundary layer | 1 rural | 2 urban | 3 oceanic | 4 tropospheric |
| Aerosol optical depth (AOD) | 0.01 | | 0.07 | 0.14 | |
| Cloud-layer optical depth ($\tau$) | 0 | 0.1 | 0.5 | 0.9 | 10 |
| Integrated ozone concentration(atm-cm) | 0.3 | | | | |
| Integrated water vapor amount ($\mathrm{g\,cm^{-2}}$) | 0.35 | | | | |
| Surface altitude (km) | 2.1 | | | | |
| Optical depth of each stratospheric aerosol layer | 0.013 | | | | |
| Atmospheric profile | 3 - Mid-Latitude Winter (AFGL standards) | | | | |

## SBDART settings

The main SBDART input parameters used in this study are the aerosol optical depth (AOD), cloud-layer optical depth ($\tau$), boundary-layer aerosol type selector (IAER) and SZA (Table 2). For the cloud properties, we used liquid water droplets with a radius of 8 μm. These parameters have been selected after a principal component analysis of the spectral absorbed energy. The principal component analysis aimed at obtaining a list of representative parameters with the most pronounced influence on the absorbed energy spectrum. For each value of the identified parameter and for each wavelength between 320 and 4000 nm, an irradiance profile is calculated.

**TARTES settings**

The main TARTES input parameters used in this study are the surface specific area (SSA) of the first layer of the snowpack, the snow water equivalent (SWE) for each layer of the snowpack and the LAPs concentration. We consider a snowpack with 3 layers of varying thickness and density (Table 3). These layers represent at most the first 20 centimeters of the snowpack, whose physical properties largely determine the albedo of the snow. The principal parameters on albedo calculation are the SSA of the first layer, the SWE of the first layers and the LAPs concentration of two first layers of the snowpack. We selected a wide range of SSA values (2 to 155 $\mathrm{m^2 kg^{-1}}$) in order to cover most of the snow types found on Earth (Domine et al., 2007). SWE gives the mass of snow and is the product between thickness ($t$) and density ($d$). For pure snow (without LAPs), the SWE values are provided for the first three layers of the snowpack. For snow with LAPs, the SWE and LAPs concentration (for soot and dust) are provided for the first two layers of the snowpack. For layer 3 and layer 4, the values of all input parameters are fixed. The ranges of LAPs content for soot and dust have been selected to cover the wide range of conditions that can be encountered in the various regions of the world from almost pristine snow in Antarctica to highly polluted snow (see e.g. Table 2 in Tuzet et al. (2020) for soot and Sterle et al. (2013) for dust). As recommended by Libois et al. (2014), we set the shape parameters $B$ and $g$ to 1.6 and 0.85. The refractive index from Warren and Brandt (2008) was used.

## 3 Results

In this section, we compare the simulated broadband absorbed energy resulting from VALHALLA for 30 $tps$ with that obtained with TARTES-SBDART for the same spectral range between 320 and 4000 nm. We first analyse the impact of incident solar radiation, cloud cover conditions and snow properties on the errors in the estimated absorbed energy and albedo. The efficiency of the method is then compared to the TARTES-SBDART calculation for different spectral resolutions ranging from 1 nm (reference simulations) to 100 nm.

### 3.1 Sensitivity of the absorbed energy to input parameters

Figure 2 shows the sensitivity of the median error on the absorbed energy calculated by the method to the atmospheric and snowpack physical properties. The calculated energy for one simulation is compared to the reference absorbed energy, calculated by TARTES-SBDART at 1 nm resolution, for the same simulation and each SZA. Overall, the median error on the broadband absorbed energy calculated for all simulations decreases with increasing SZA. Concerning the atmospheric properties, the median error on absorbed energy exhibits a stronger sensitivity to $\tau$ than to AOD. The median errors are small for values of $\tau$ equal to 0.1, 0.5 and 10 (absolute difference < 1 $\mathrm{W\,m^{-2}}$) but remain larger for values equal to 0.0, 0.9 and 5.0 (e.g. median error = 3.6 $\mathrm{W\,m^{-2}}$ for $\tau$ = 5 and SZA = 10°). Errors are lower when using an adequate reference irradiance profile ($\tau$ of simulation ($\tau_{simu}$) equal to $\tau$ of reference ($\tau_{ref}$), $\tau$ = 0.5 and 10) and the calculated absorbed energy is therefore very sensitive to $\tau$ (median errors between 1.5 and -3.6 $\mathrm{W\,m^{-2}}$). Regarding AOD, the median errors are small (absolute difference < 0.5 $\mathrm{W\,m^{-2}}$) and show little changes with $\tau$. This demonstrates that AOD exerts a very small influence on the median error

**Table 3.** Snow properties of simulations. The spectral albedo is calculated for eight values of SZA and five values of SSA for the snowpack first layer. For snow without LAPs, the SWE values are provided for the first three layers of the snowpack. For snow with LAPs, the SWE and the light absorbing particles concentration (for soot and dust) are provided for the first two layers of the snowpack. For layer 3, the values of all input parameters, besides soot and dust contents, are constant. For layer 4, all input parameters are constant.

| | Solar zenith angle (SZA, °) | 10 ; 20 ; 30 ; 40 ; 50 ; 60 ; 70 ; 80 | | | |
|---|---|---|---|---|---|
| **Layer 1** | SSA ($m^2$ $kg^{-1}$) | 2, 5, 42, 82, 155 | | | |
| | SWE (kg $m^{-2}$) | 1 | 4 | 15 | |
| | Thickness ($t$, m) | $t : 0.01$ | $t : 0.02$ | $t : 0.05$ | |
| | Density ($d$, kg $m^3$) | $d : 100$ | $d : 200$ | $d : 300$ | |
| | Soot concentration (ng $g^{-1}$) | 0, 100, 200 | | | |
| | Dust concentration (ng $g^{-1}$) | 0, 25000, 50000 | | | |
| **Layer 2** | SSA ($m^2$ $kg^{-1}$) | 42 | | | |
| | SWE (kg $m^{-2}$) | 1 | 4 | 15 | |
| | Thickness ($t$, m) | $t : 0.01$ | $t : 0.02$ | $t : 0.05$ | |
| | Density ($d$, kg $m^3$) | $d : 100$ | $d : 200$ | $d : 300$ | |
| | Soot concentration (ng $g^{-1}$) | 0, 100, 200 | | | |
| | Dust concentration (ng $g^{-1}$) | 0, 25000, 50000 | | | |
| **Layer 3** | SSA ($m^2$ $kg^{-1}$) | 42 | | | |
| | SWE (kg $m^{-2}$) | 1 | 4 | 12.5 | 15 |
| | Thickness ($t$, m) | $t : 0.01$ | $t : 0.02$ | $t : 0.05$ | $t : 0.05$ |
| | Density ($d$, kg $m^3$) | $d : 100$ | $d : 200$ | $d : 250$ | $d : 300$ |
| | Soot concentration (ng $g^{-1}$) | 0, 100 | | | |
| | Dust concentration (ng $g^{-1}$) | 0, 25000 | | | |
| **Layer 4** | SSA ($m^2$ $kg^{-1}$) | 42 | | | |
| | SWE (kg $m^{-2}$) | 600 | | | |
| | Thickness ($t$, m) | $t : 2$ | | | |
| | Density ($d$, kg $m^3$) | $d : 300$ | | | |
| | Soot concentration (ng $g^{-1}$) | 0 | | | |
| | Dust concentration (ng $g^{-1}$) | 0 | | | |

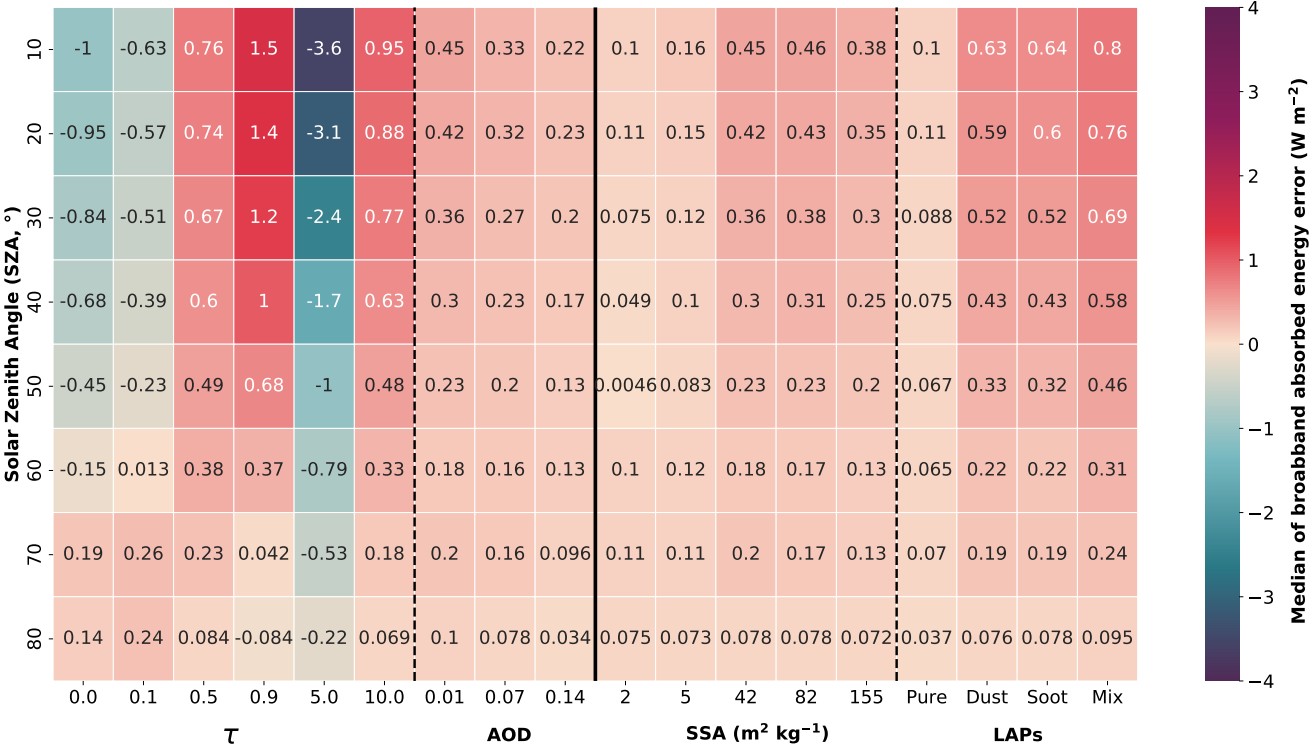

**Figure 2.** Median error of the broadband absorbed energy for varying SZA, the $\tau$, the AOD, the SSA m$^2$ kg$^{-1}$ and LAPs type. The median error on the broadband absorbed energy is calculated on the ensemble of all the simulations described in Section 2.2.3 using 30 $tps$ using TARTES-SBDART at 1 nm resolution as reference.

and thus on the calculation of the energy absorbed by the method. Concerning the properties of the snow cover, the SSA value of the first layer has little impact on the error of the absorbed energy. For the different SSA values, the median errors are small (absolute difference $< 0.5$ W m$^{-2}$) and vary little depending on the value studied. The presence of LAPs in the snowpack leads to an increase in the median error (absolute difference $< 1$ W m$^{-2}$) compared to pure snow (absolute difference $< 0.1$ W m$^{-2}$). Overall the method slightly overestimates the energy absorbed by the snowpack (mostly positive errors). The error is not very sensitive to the physical properties of the snowpack and to the AOD. However, the error is very sensitive to $\tau$ of the simulations and thus to the $\tau$ chosen for the reference profile. The sensitivity to cloud conditions is investigated in more details in the next section.

### 3.2 Sensitivity to cloud cover conditions

Figure 2 shows the median errors on the broadband absorbed energy for all the simulations. For each of them, the bias on the broadband absorbed energy is shown in Fig 3. Figure 3a,b show the distribution of the biases in the broadband absorbed

energy and the broadband albedo as a function of SZA and $\tau$. These biases are determined as the difference between the absorbed energy calculated by VALHALLA and TARTES-SBDART at 1 nm resolution. Overall,the broadband albedo biases vary little with SZA and the biases of the absorbed energy decrease with SZA. This is consistent with higher absorbed energy for lower SZA (higher incoming radiation and lower albedo). For simulations with a value of $\tau$ equal to 0.5 and 10 and each SZA value, an adequate reference irradiance profile is used ($\tau_{simu} = \tau_{ref}$). More than 75% of the errors are positive, meaning that VALHALLA overestimates the absorbed energy. The errors are low and range between -1 and 1.5 $\mathrm{W\,m^{-2}}$ with a median error of -0.76 and -0.95 $\mathrm{W\,m^{-2}}$ for $\tau$ equal to 0.5 and 10, respectively. For the simulations with a value of $\tau$ equal to 0, 0.1 and 0.9, the reference irradiance profile used has a $\tau$ value different from the simulations ($\tau_{simu} \neq \tau_{ref}$). For those with $\tau$ equal to 0 and 0.1, the biases are overall negative (for approximately 90% of the biases) and varies between -4 and 0.5 $\mathrm{W\,m^{-2}}$. For $\tau$ equal to 0.9, the biases of the absorbed energy are positive (for more than 95% of biases) and range between -0.5 and 2.8 $\mathrm{W\,m^{-2}}$. The absolute error in the absorbed energy is decreasing with SZA for all $\tau$ values.

Figure 3c,d show the spectral variation of the reference absorbed energy calculated by TARTES-SBDART and that calculated by VALHALLA. The absorbed energy profiles presented are calculated for a simulation with two values of $\tau$ (0 and 10) between 320 and 4000 nm at 1 nm resolution. The spectral error of the absorbed energy is also calculated as the difference between the energy calculated by VALHALLA and TARTES-SBDART. For the simulation with $\tau$ equal to 0 (clear sky, Fig 3c), the majority of the errors are negative and are up to -9 $\mathrm{W\,m^{-2}\,\mu m^{-1}}$). The higher errors are located at the wavelengths where the absorption is the highest (between 1 and 1.5 µm). For the one with $\tau$ equal to 10 (full overcast, Fig 3d), the method represents very well the absorbed energy (errors close to 0 $\mathrm{W\,m^{-2}\,\mu m^{-1}}$). The use of an adequate reference irradiance profile ($\tau_{simu} = \tau_{ref}$) thus leads to a decrease of the error on the spectral and broadband absorbed energy, despite a slight overestimation of the energy absorbed by the method (positives errors). However, when $\tau_{simu} \neq \tau_{ref}$, the error on the absorbed energy is higher. When $\tau$ of the simulation is lower than $\tau$ of the reference profile ($\tau_{simu} < \tau_{ref}$), the absorbed energy is underestimated by the method (globally negative errors). When $\tau$ of the simulation is higher than $\tau$ of the reference profile $\tau_{simu} > \tau_{ref}$, the absorbed energy is overestimated by the method (positives errors). The biases of the absorbed energy are therefore very sensitive to $\tau$ of the simulations and therefore to the optical thickness chosen for the reference profile.

### 3.3 Sensitivity to snow physical properties

Figure 4a,b. show the distribution of the biases in broadband absorbed energy and albedo for varying SZA and SSA of the first layer of the snowpack. Broadband energy biases decrease with increasing SSA as the absolute absorbed energy is also decreasing. For an SSA equal to 2 $\mathrm{m^2 kg^{-1}}$, the biases vary between -2 and 4 $\mathrm{W\,m^{-2}}$ as opposed to a variation of -1.5 to 1.5 $\mathrm{W\,m^{-2}}$ for an SSA equal to 155 $\mathrm{m^2 kg^{-1}}$.

Figure 4c,d. show the spectral variation of the reference absorbed energy calculated by TARTES-SBDART and that calculated by the method. The absorbed energy profiles presented are calculated for a simulation with two extreme SSA values (5, representative of old snow and 155 $\mathrm{m^2\,kg^{-1}}$ for new snow) between 320 and 4000 nm at 1 nm resolution. For these two simulations, the spectral errors of the absorbed energy are greater for an SSA value equal to 5 $\mathrm{m^2\,kg^{-1}}$ (up to -10 $\mathrm{W\,m^{-2}\,\mu m^{-1}}$), than for an SSA of 155 $\mathrm{m^2\,kg^{-1}}$ (> -8 $\mathrm{W\,m^{-2}\,\mu m^{-1}}$). The highest errors for these two simulations are located at the wavelengths

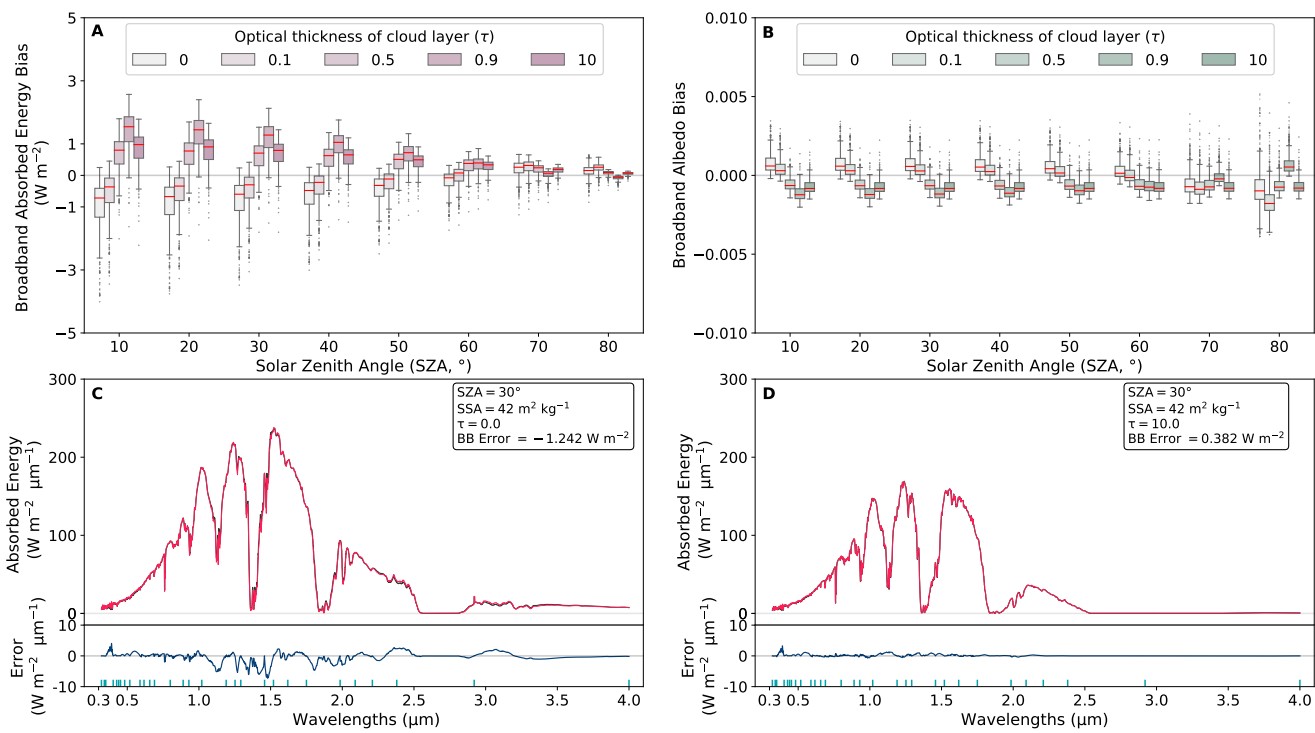

**Figure 3.** Bias on the broadband absorbed energy (A) and on the broadband albedo (B) as a function of the SZA and $\tau$. The biases are calculated for all the simulations described in section 2.2.3, between the absorbed energy calculated by the method and the reference absorbed energy calculated by TARTES-SBDART. The red lines indicate the median (same as in Fig 2), the box shows the 25th to 75th percentiles and the whiskers show the 5th to 95th percentiles. Example of absorbed energy profiles by a snowpack without LAPs as a function of wavelength calculated for an SZA of 30° and a $\tau$ value of 0 (clear-sky, C) and a $\tau$ value of 10 (full overcast, D). The black lines represent the absorbed energy calculated by TARTES-SBDART at 1 nm resolution, and the red lines represent the absorbed energy calculated by VALHALLA. In blue, the errors on the absorbed energy for these same simulations as a function of wavelength. The green vertical lines represent the $tps$ used in VALHALLA. BB error correspond to the broadband error for the absorbed energy in panels C and D.

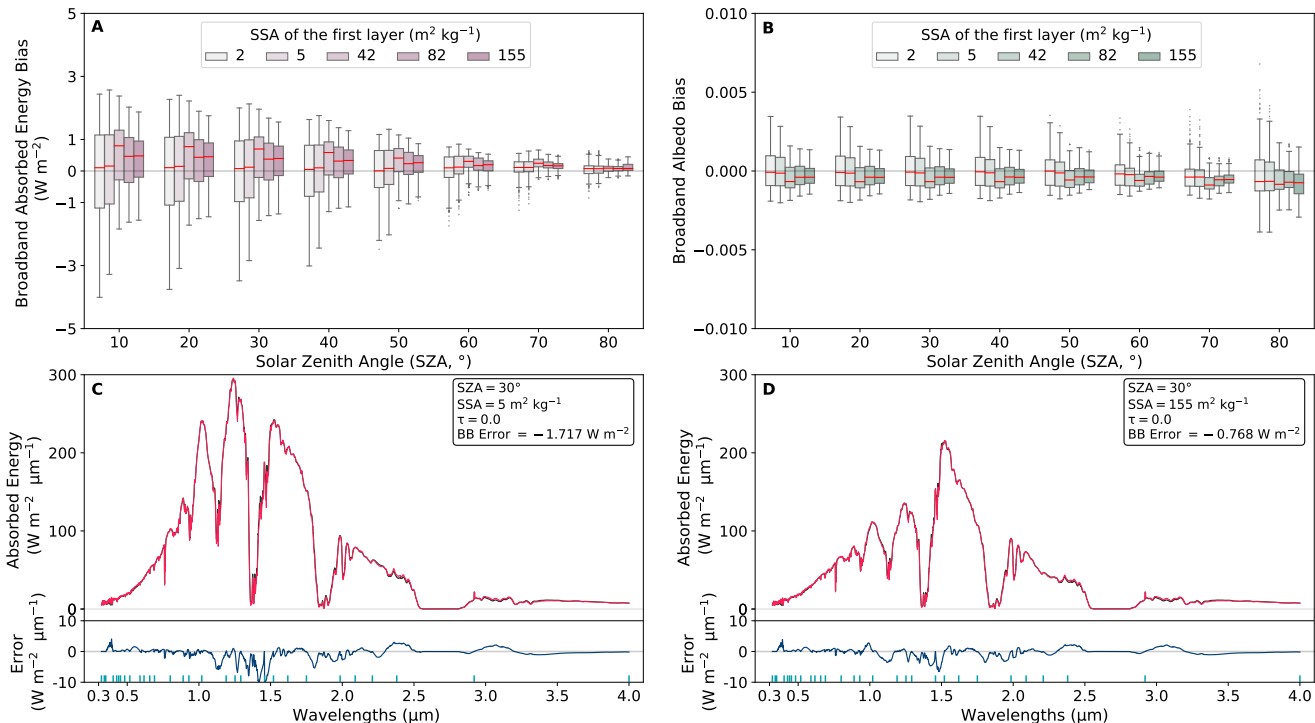

**Figure 4.** Bias on the broadband absorbed energy (A) and on the broadband albedo (B) as a function of the SZA and the SSA of the first layer ($m^2 kg^{-1}$). The biases are calculated for all the simulations described in sections 2.5.3. and 2.5.4., between the absorbed energy calculated by the method and the reference absorbed energy calculated by the TARTES-SBDART. The red lines indicate the median, the box shows the 25th to 75th percentiles and the whiskers show the 5th to 95th percentiles. Example of absorbed energy profiles by a snowpack without LAPs as a function of wavelength calculated for an SZA of 30° and an SSA value of 5 $m^2 kg^{-1}$ (old snow, C) and an SSA value of 155 $m^2 kg^{-1}$ (fresh snow, D). The black lines represent the absorbed energy calculated by TARTES-SBDART at 1 nm resolution, and the red lines represent the absorbed energy calculated by VALHALLA. In blue, the errors on the absorbed energy for these same simulations as a function of wavelength. The green vertical lines represent the $tps$ used in VALHALLA.

where absorption is maximal (between 1 and 1.5 μm). When the snowpack is absorbing a large amount of energy, such as for low SSA, the biases on the spectral and broadband absorbed energy increase. The biases on the absorbed energy are therefore relatively sensitive to the SSA of the first layer of the snowpack and thus remain very sensitive to the absorbing properties of the snowpack.

## 3.4 Sensitivity to LAPs

Figure 5a,b. show the distribution of the biases in broadband absorbed energy and albedo for various SZA and LAPs contents. Broadband energy biases increase with the presence of LAPs in the snowpack. However, for pure snow, the biases are more

negative than for snow with LAPs and the spread of the biases is greater (between -4 and 1.2 $\mathrm{W\,m^{-2}}$ ). For snow with dust or soot, the distribution of biases is very similar (between -1 and 2 $\mathrm{W\,m^{-2}}$) whereas for snow with a mix of dust and soot, the spread is larger (between -2 and 2.7 $\mathrm{W\,m^{-2}}$).

Figure 5c,d. show the spectral variation of the reference absorbed energy calculated by TARTES-SBDART and that calcu-
lated by VALHALLA. The absorbed energy profiles presented are calculated for a simulation with two LAPs concentration contained in the snowpack (a snowpack which contains 25000 $\mathrm{ng\,g^{-1}}$ of dust and a snowpack which contains a mix of 100 $\mathrm{ng\,g^{-1}}$ of soot and 50000 $\mathrm{ng\,g^{-1}}$ of dust) between 320 and 4000 nm at 1 nm resolution. LAPs being highly absorbent at the beginning of the spectrum (between 0.3 and 0.8 μm, Warren, 1982), the highest spectral absolute errors are consequently located in this wavelength range. The method is indeed based on the ice refractive index (e.g. Eq. 9) and thus partly failed to
reproduce changes in the refractive index due to the presence of LAPs. For a snowpack containing a mix of LAPs (5d), the errors at the beginning of the spectrum are higher than for a snowpack containing only dust (5c). The presence of a mix of LAPs in the snowpack generates errors of up to -30 $\mathrm{W\,m^{-2}\,μm}$ against maximum errors of -20 $\mathrm{W\,m^{-2}\,μm}$ for the snowpack containing only dust. The biases on the spectral and broadband energy increase with the amount of energy absorbed by the snowpack. The biases on the absorbed energy are therefore very sensitive to the LAPs content in the snowpack and thus remain
very sensitive to the absorbing properties of the snowpack.

### 3.5  Comparison to calculations with regular spectral resolution

In Fig. 6 we compared the broadband albedo bias obtained with the VALHALLA methods to the bias obtained for spectral resolution varying from 2 to 100 nm. The comparison was performed using the simulations from section 2.2.3. For regular spectral resolution, the absolute bias generally increases with the spectral steps and tends to be more negative. This means that
the bias on the absorbed energy tends to be more positive when the spectral steps increase. We believe that for low spectral resolution, the integration over the spectrum is missing the absorption bands leading the integral to be higher than for smaller spectral steps (see e.g. the spectrum in Figs. 3-5). The VALHALLA method presents biases on the broadband albedo with absolute difference lower than 0.005 which are comparable to the bias obtained with resolutions lower or equal than 20 nm (reference resolution used at MétéoFrance in research activities). The VALHALLA method uses 30 wavelengths (30 $tps$)
when the calculation at 20 nm resolution requires 184 wavelengths. Thus, for the same bias on the broadband albedo, the VALHALLA method uses six times fewer bands than a calculation at 20 nm resolution.

### 4  Discussions

We presented the VALHALLA method for calculating absorbed energy and albedo based on a calculation of the main variables explaining the variations in absorbed energy using spectrally fixed radiative variables. We determined 30 $tps$, corresponding
to the local minima and maxima of the absorbed energy at which the exact calculation of the absorbed energy is performed. In addition, we used 16 different reference irradiance profiles to interpolate between these $tps$. We evaluated the accuracy of the method for several atmospheric and snow properties that influence the amount of energy reaching the ground and snow albedo,

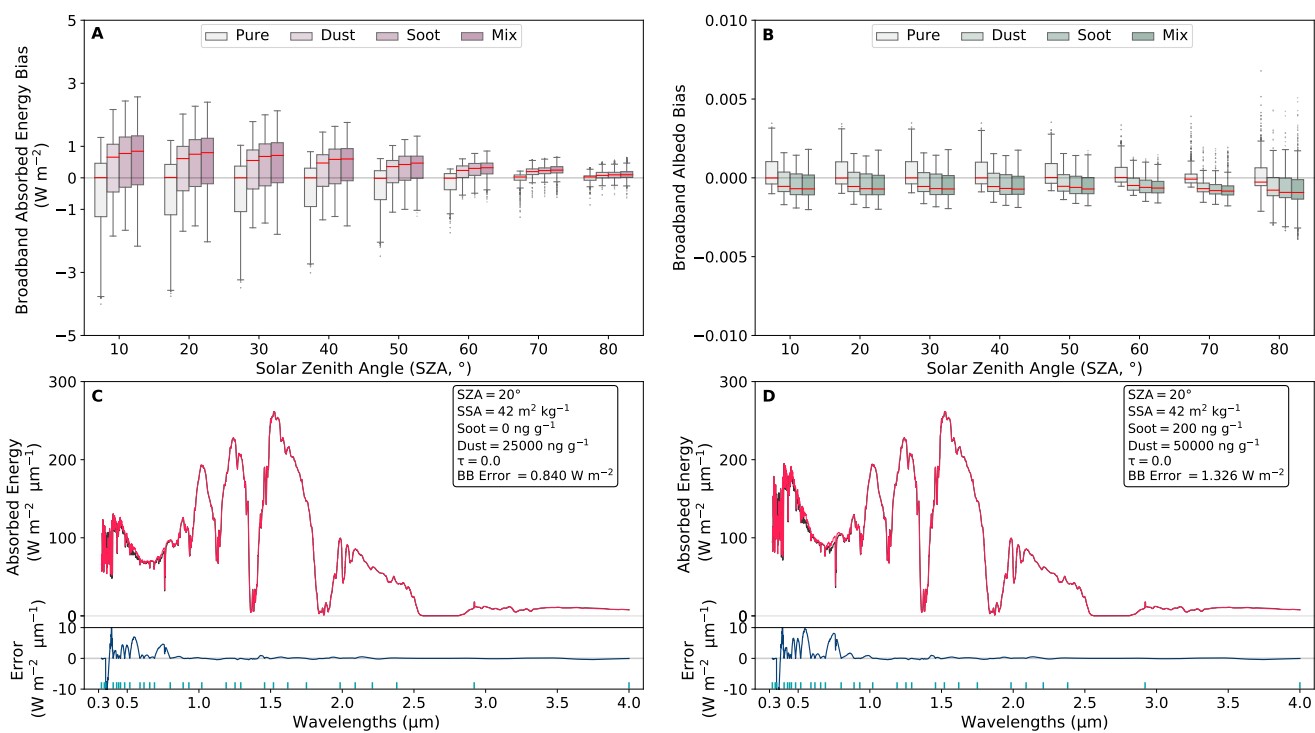

**Figure 5.** Bias on the broadband absorbed energy (A) and on the broadband albedo (B) as a function of the SZA and the LAPs content. The biases are calculated for all the simulations described in sections 2.5.3. and 2.5.4., between the absorbed energy calculated by the method and the reference absorbed energy calculated by TARTES-SBDART. The red lines indicate the median, the box shows the 25th to 75th percentiles and the whiskers show the 5th to 95th percentiles. Example of absorbed energy profiles by a snowpack with LAPs as a function of wavelength calculated for an SZA of 30° for a snowpack which contains dust (C) and a mix of soot and dust (D). The black lines represent the absorbed energy calculated by TARTES-SBDART at 1 nm resolution, and the red lines represent the absorbed energy calculated by VALHALLA. In blue, the errors on the absorbed energy for these same simulations as a function of wavelength. The green vertical lines represent the *tps* used in VALHALLA.

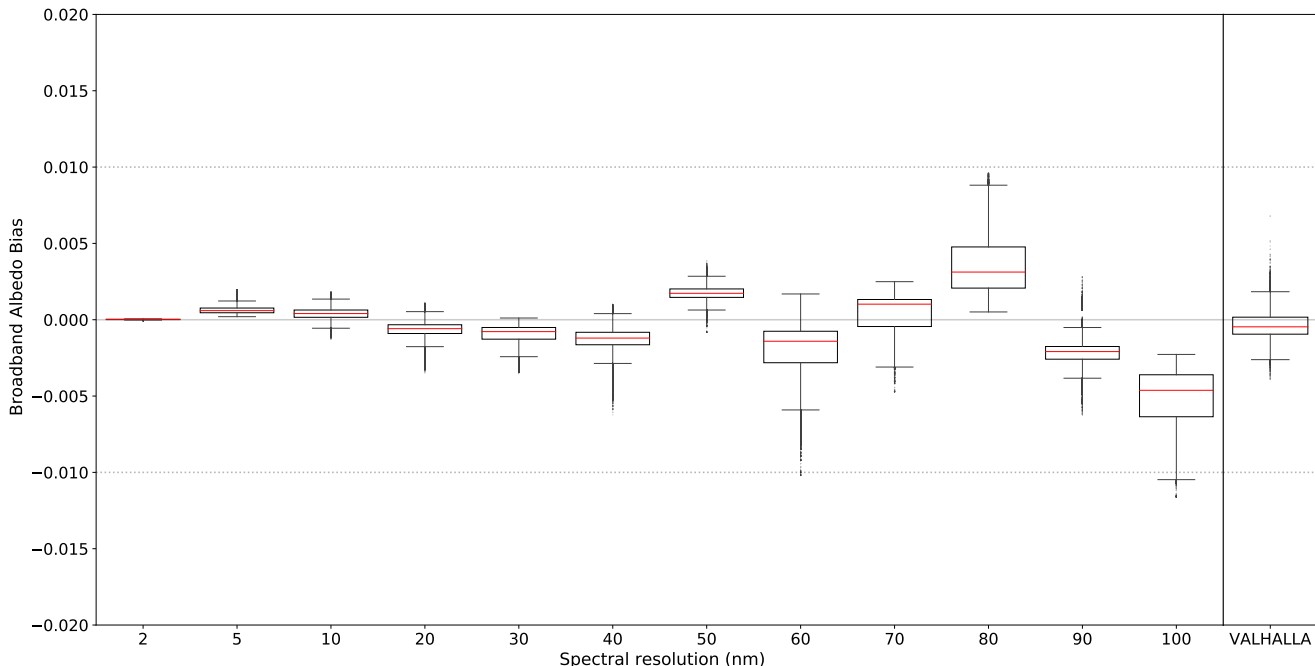

**Figure 6.** Errors on the broadband albedo for different constant spectral resolutions (left) and comparison with the errors of our method (right). For these resolutions, the broadband albedo is computed by TARTES-SBDART and is compared to the one computed at 1 nm resolution. The red lines indicate the median (same as in Fig 2), the box shows the 25th to 75th percentiles and the whiskers show the 5th to 95th percentiles. The grey dotted lines correspond to an error of $\pm 0.01$ as in Gardner and Sharp (2010).

such as $\tau$, AOD, SSA and LAPs content. We have shown that absorbed energy and albedo errors due to the use of this method are small (absolute difference $< 1\ \mathrm{W\,m^{-2}}$ for broadband absorbed energy and absolute difference $< 0.005$ for broadband
albedo) and correspond to a factor 6 in terms of computation times compared to calculations made at 20 nm resolution.

### 4.1 On the accuracy of the method

We have shown that the absorbed energy calculated by VALHALLA is very sensitive to $\tau$ of the simulation and therefore to the use of an adequate reference irradiance profile. The use of a reference profile that is not adapted to the irradiance of the simulation ($\tau_{simu} \neq \tau_{ref}$) leads to a clear increase in the error on the absorbed energy. To reduce the uncertainties
resulting from the method, a preliminary calculation of reference irradiance profiles adapted to each cloud condition could be initialised. The reference profiles can also be adapted to the cloud types (liquid water or ice droplets, droplets radius) when this information is available together with the solar radiation. Therefore, for all optical thickness values used in the simulations, the irradiance in the method can be adapted. The presence of LAPs in the snow cover leads to an increase in errors on the absorbed energy, especially at the beginning of the spectrum where LAPs strongly impact the absorption efficiency. The method fails

to accurately represent the absorbed energy between two $tps$ in the visible range in presence of LAPs since it is based on the ice refractive index only. To reduce the uncertainties at the beginning of the spectrum and thus reduce the broadband error, it would be possible to increase the number of $tps$ at the beginning of the spectrum. However, this would increase the calculation time. The choice of the number of $tps$ is discussed later in this section. The other variables studied, such as the SZA and the SSA, appear to be less influential. The associated absolute error evolves as a function of the amount of energy absorbed by the snowpack and is therefore driven by the absorbing factors such as the SZA and the SSA. The error on the absorbed energy, therefore, increases with a decrease in the solar angle and a decrease in the SSA value of the first layer of the snowpack. Although the absolute error decreases with SZA, the relative error generally increases for high SZA, as can be seen in Figs. 3-5. For SZA higher than 85° (not tested here), the broadband albedo might be interpolated between the value at 85° and 1. The choice of an adequate reference irradiance profile for the simulation globally determines the accuracy of the absorbed energy error calculated by VALHALLA. However, the choice of $tps$ is also a determining factor in a good estimate of the energy absorbed by the method.

## 4.2 Sensitivity to tie points

The accuracy of the method is sensitive to the locations and to the number of $tps$. An increase or a decrease in the number of $tps$ could improve or alter the representation of the absorbed energy. Using a too large number of $tps$ leads to a decrease in the calculated error but increases the calculation time, especially if the $tp$ number is increased at the beginning of the spectrum to compensate for the oscillations of the absorbed energy when the snowpack contains LAPs. For a lower $tp$ number, the oscillations at the beginning of the spectrum due to LAPs are not well represented by the method and this leads to a significant increase in the error. With the use of 15 $tps$, the error on the broadband albedo increases globally by a factor of 10 to 15 for snow containing LAPs. With 10 $tps$, the error increases by a factor of 25 and 50 for the same type of snow. The effect of LAPs on the absorbed energy is therefore poorly represented when the number of $tp$ is too low. The use of 30 $tp$ is, therefore, a good compromise between precision for snowpacks containing LAPs and calculation time.

## 4.3 Comparison to other existing methods

Gardner and Sharp (2010) developed an snow broadband albedo parameterization accounting for changes in the snow and atmospheric properties. The computational cost of such albedo parameterization is very small and the accuracy is around 0.01 for the broadband albedo (compared to reference calculations at 10 nm resolution). This accuracy is depicted in Fig. 6 by the grey dotted horizontal lines. The accuracy of VALHALLA is roughly an order of magnitude lower. However, the albedo parameterization of Gardner and Sharp (2010) and VALHALLA fulfil two different goals since VALHALLA requires accurate snow and atmosphere radiative transfer calculation for the $tps$. The computational cost of Gardner and Sharp (2010) is thus lower than the one of VALHALLA.

The SNOWBAL coupling scheme from Van Dalum et al. (2019) described in the introduction of our study provides albedo calculation with an accuracy better than 0.01. Thanks to the physics of the snow radiative transfer model TARTES, SNOWBAL accurately calculates the vertical distribution of the absorbed energy in the different snow layers. For VALHALLA when

using the method with TARTES or with any other multilayer radiative transfer model for snow (e.g. SNICAR, He et al., 2018), the vertical distribution of the absorbed energy is calculated for the $tps$ but the vertical profile of broadband absorbed energy is not directly available. This would require further development of the method. SNOWBAL used 14 representative wavelengths (RWs) for which accurate snow radiative transfer calculations are performed. The number of RWs depend on the number of narrowbands available for the solar radiation. For VAHLALLA, the accurate snow and atmosphere radiative transfer simulations are performed for 30 $tps$. When using 15 $tps$, our method fails to converge to a good representation of the broadband albedo (increasing the error by a factor of 10 to 15). The use of more $tps$ (30) is therefore necessary for an improved representation of the broadband albedo. $tps$ and RWs are not directly comparable, since the number of RWs depends on the number of narrowbands available and it is not the case of the number of $tps$.

VALHALLA and SNOWBAL fulfill two different niches. SNOWBAL indeed required accurate snow radiative transfer calculation and accurate atmospheric conditions (cloud water content, direct-to-diffuse irradiance radiation, ...). VALHALLA requires both snow and atmophere radiative transfer calculations for the $tps$. This difference together with the need for more than 15 $tps$ implicates that the computational cost of VALHALLA is higher than the computational cost of SNOWBAL. However, the accuracy of the SNOWBAL methods depend on the number and range of the narrowband solar radiation available. SNOWBAL accuracy increases when the sub-band spectral variability is reduced. Here, we used the VALHALLA method with broadband solar irradiance inputs, i.e. the worst case. The method was also tested with narrowband solar radiation inputs (from AROME (Seity et al., 2011), not shown) providing similar accuracy on the absorbed energy than the one presented with broadband inputs.

## 4.4 Implementation considerations

The VALHALLA method has been developed to provide accurate calculation of the solar energy absorbed by the snowpack at low computational cost compared to full spectral calculation. The VALHALLA method requires accurate calculation of the spectral absorbed energy for the $tps$. In the study, this is based on TARTES and SBDART models but any other radiative model could be used (e.g. SNICAR for snow (He et al., 2018), Bird and Riordan (1986) for the atmosphere). The overall accuracy of the calculation depends on the choice of the radiatiave transfer model for snow and for the atmosphere. We believe that the VALHALLA method is an especially efficient compromise between accuracy and computational cost when only broadband (or large narrowbands) solar irradiance value are available from the atmospheric model. This is the case for example for the detailed snowpack model Crocus in the land surface model SURFEX (Tuzet et al., 2017), this is also the case when surface simulations are performed offline (not coupled), i.e. using atmospheric reanalysis or measurements as inputs.

## 5 Conclusions

In climate models, energy fluxes are most often given for narrow and large spectral bands. The low spectral resolution of these fluxes therefore leads to uncertainties in the determination of radiative variables such as snow albedo that are key for energy exchanges at the surface. This study presents a new method VALHALLA for calculating the spectral albedo of snow

based on the determination of key atmospheric and snow variables explaining variations in absorbed energy using spectrally fixed variables. For this method, tie points ($tps$) and reference irradiance profiles are calculated to incorporate the absorbed energy and the reference irradiance. The absorbed energy is then interpolated for each wavelength present between two $tps$ with adequate kernel functions derived from radiative transfer theory for snow and the atmosphere.

For the different properties of the atmosphere and snow studied, the cloud-layer optical depth ($\tau$) and the LAP content of the snow cover are the main variables influencing the calculation of the absorbed energy by the method. Indeed, when the value of $\tau$ of the simulation is equal to that of the reference irradiance profile, the method converges towards a value of absorbed energy close to that calculated as a reference. On the other hand, when this value is not equal to that of the reference profile, differences in absorbed energy are noticeable at certain wavelengths. For snowpacks containing LAPs, the method encounters difficulties in representing the variation in absorbed energy at the beginning of the spectrum and therefore generates significant differences in energy. The use of reference profiles with an adequate value of SZA is necessary to the good accuracy of the method.

The VALHALLA method, therefore, determines the absorbed energy for all wavelengths between 320 and 4000 nm using 30 $tps$. This number of $tps$ is necessary for a good representation of the absorbed energy when the snow contains LAPs. Despite an overestimation of the energy absorbed by the method, the results obtained with 30 $tps$ are similar to the results of a TARTES-SBDART at 20 nm. This results in a reduction of the calculation time by a factor of 6 (30 $tps$ versus 180 wavelengths). In addition to the performance in calculation time, the method is versatile and adaptable to any atmospheric input (broadband, narrowband).

In conclusion, the development of the method VALHALLA presented here allows a considerable reduction in calculation time while maintaining a good representation of the spectral albedo. One of the perspectives would be to integrate this method in a radiative scheme of a global or regional climate model in order to drastically reduce the calculation time and to largely improve the albedo calculation compared to more common broadband and/or narrowband calculations.

*Code and data availability.* The VALHALLA v1.0 development and data presented and described in this article (Veillon et al., 2020) is available for download at https://doi.org/10.5281/zenodo.4570565.

TARTES is freely avaibable on the website: http://pp.ige-grenoble.fr/pageperso/picardgh/tartes/. SBDART is freely available on the website: https://github.com/paulricchiazzi/SBDART.

*Author contributions.* FV, MD and MF started this project and developped the method. FV ran the simulations and wrote the first draft of the manuscript. FV, MD and CA performed the analysis . All authors discussed and revised the manuscript.

*Competing interests.* The authors declare that they have no conflict of interests.

*Acknowledgements.* CNRM/CEN is part of Labex OSUG@2020 (ANR-10-LABX-0056). This work was partly funded by CNES APR
MIOSOTIS and by ANR grant EBONI (ANR-16-CE01-0006). M.D. has received funding from the European Research Council (ERC)
under the European Union's Horizon 2020 research and innovation programme (grant agreement No 949516, IVORI). The authors are
grateful to Ghislain Picard and Quentin Libois for discussion on the VALHALLA method. The authors are also grateful to the two referees
Joseph Cook and Christiaan Van Dalum for helpful and relevant comments on the manuscript.

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
