# Peer review of "A versatile method for computing optimized snow albedo from spectrally fixed radiative variables: VALHALLA v1.0"

_Geoscientific Model Development, 2020_

## Author Comment (AC1)

Reviewer comments are in black. Author response in blue and proposed changes in the manuscript in **bold blue or in latex fonts**. Page and line numbers refer to the first version of the manuscript.

**Comments by Joseph Cook**

This paper aimed to describe a new method for estimating full-resolution spectral albedo from calculation at a subset of reference wavelengths. The rationale for this is that models with lower numerical load than full RTMs are required for regional climate models. Current models that do this are subject to biases because the regimes used to interpolate between reference wavelengths lead to biases.

In my estimation, the paper succeeds in demonstrating the new algorithm and the subject matter is well within the scope of GMD. Overall, they have clearly described their method, provided a transparent report of its performance relative to TARTES and identified an optimal configuration that balances computation time and accuracy. Therefore, I support this manuscript being published in GMD.

AC: The authors are very thankful to Joseph Cook for the time spent on reviewing the manuscript and for the positive feedback. Point by point answers are provided below along with proposed changes in the manuscript.

The areas that I think could be improved are:

a) it took me a few reads to really understand what benefit the new model provides to the community – I think just reworking the introduction slightly to make it crystal clear why this is useful might be helpful.

AC: We agree that this was not sufficiently clear in the first version of the manuscript. In the revised version the end of the introduction was fully rewritten and two sections were added in the discussion to discuss the pros and cons of VALHALLA compared to other existing methods. The modifications proposed are reported below :

Introduction

[revised manuscript text omitted]

b) the comparison with the 14 tps model used by van Dalum et al. (2019) was very informative. Given that the 15 tps version of VALHALLA failed to give a good representation of the albedo, and presumably there is a computational cost associated with adding tps, can you clarify the argument for using VALHALLA in its 30 tps form in a regional climate model in preference to SNOWBAL?

AC: A discussion on the pros and cons of the different methods (including VALHALLA and SNOWBAL) has been added in the discussion (see proposed text in the response to your comment a) above). Please see also answers to Christiaan Van Dallum general comments 1, 2 and 3.

c) Is there a physical explanation for the relationship between model bias and SZA/SSA?

AC: Yes there is a physical explanation. Figures 3,4 and 5 show that the error in absorbed energy is increasing with decreasing SSA and decreasing SZA. This is due to the fact that the higher absorbed energy is found for low SSA (lower albedo) and for low SZA (lower albedo + higher incoming solar energy). This was explained for SSA p13 lines 236-237. This was also explained for impurities p14 lines 253-254, but the explanation was missing for SZA.

In the new version of the manuscript, this was added p 8 L185-186 : "Overall,**the broadband albedo biases vary little with SZA and the biases of the absorbed energy decrease with SZA. This is consistent with higher absorbed energy for lower SZA (higher incoming radiation and lower albedo).**"

d) Can you give any more detail about the "systematic error" at 400 nm? This seems like it could be a significant issue, but is not explained in much detail in the manuscript. Is this the same as what is referred to in the discussion lines 283-285?

AC: Yes this is the same as what is discussed in lines 283-285. This error appears in the presence of light absorbing particles. Since the method is based on the refractive index of ice, e.g. Eq. (10) in the paper, the interpolation is not really successful when the refractive index of another material is in play (e.g. snow with light absorbing particles in the visible wavelengths). As a consequence, adding more tie points in the visible helps but does not fully remove the errors (Fig. 5c,d).

 This is now detailed P14  line 250:

"LAPs being highly absorbent at the beginning of the spectrum (between 0.3 and 0.8 µm, Warren, 1982), the most important errors are **consequently** located in this wavelength range. **The method is indeed based on the ice refractive index (e.g. Eq. 10) and thus partly failed to reproduce changes in the refractive index due to the presence of LAPs.** "

and in the discussion P16-17 L282-287 :

"The presence of LAPs in the snow cover leads to an increase in errors on the absorbed energy, **especially at the beginning of the spectrum where LAPs strongly impact the absorption efficiency**. **The method fails to accurately represent the absorbed energy between two tps in the visible range in presence of LAPs since it is based on the ice refractive index only**. To reduce the uncertainties at the beginning of the spectrum and thus reduce the broadband error, it would be possible to increase the number of tps at the beginning of the spectrum. However, this would increase the calculation time."

e) The zenodo archive really doesn't contain much helpful documentation. A quick review of the code indicates there are significant dependencies including a development environment that includes both tartes and sbdart with specific configurations – it also seems to be OS specific judging by calls out to the sbdart command line tool. I think these and related issues need to be explained in the model documentation in the form of some basic user instructions.

AC: Documentation of the archive has been improved by adding in each folder a README file that provides information on the content of each file including headers and units. Information on how to use the code with examples of running commands are given in the README file of the main folder. The whole environment has been developed under linux and this is now also specified in the README file of the main folder. Since sbdart is a .exe which can be called with python or other and tartes is a python module, only the calls out to the sbdart and tartes command lines tool are OS specific. The updated archive can be found at

:https://zenodo.org/record/5289201#.YSjk35w6_mE

---

## Author Comment (AC3)

Reviewer comments are in black. Author response in blue and proposed changes in the manuscript in **bold blue** or in latex fonts. Page and line numbers refer to the first version of the manuscript.

This study presents a new and numerically fast way to determine the snow albedo for use in climate models. The spectral albedo model TARTES together with SBDART are used to determine the energy absorbed for various snowpacks. By using kernel functions, the absorbed energy can be interpolated between tie points, which allows the absorbed energy to be calculated for a wide range of wavelengths. The authors also investigate the impact of various processes and find that the optical depth and LAP content are the most important variables.

The algorithm that the authors present is a clever way to determine the absorption of energy in snow and is also within the scope of this journal. However, I had to read the manuscript several times to properly understand the method and there are still several parts not entirely clear to me. Although VALHALLA is potentially a useful model for the community, the manuscript does not provide an adequate description on how to implement it, hindering the actual implementation in climate models that may be done by others. Consequently, I think that some parts of the manuscript should be reformulated and/or expanded on. The following comments should help with solving most issues, with, for example, P1 L1 meaning page 1, line 1.

AC: The authors are very grateful to Christiaan Van Dallum for the time spent on reviewing the manuscript and for the very comprehensive and relevant comments. Point by point answers are provided below along with proposed changes in the manuscript.

General comments

1. The most pressing concern is that the method section is hard to follow and misses some information in my opinion. As this is a vital part of the manuscript, it is hard to interpret the results if the methods are not clearly defined. For example, it took a while before I realized that VALHALLA models absorption and not albedo. I especially miss a part about how the implementation would look like in a climate model. I would suggest to rewrite or restructure most of the method section to take away the confusion. The authors may use the following structure, breaking Sect. 2 in 3 parts:

   2.1) Existing models

   2.1.1) TARTES,

   2.1.2) SBDART

   2.2) VALHALLA description

2.2.1) Theoretical considerations (currently 2.3). Inlcuding a) a discussion why a tie-point method is a clever way to go. b) Explain why the authors choose to model the absorption curve and not the spectral albedo curve.

2.2.2) Interpolation method (currently 2.4). Furthermore, some steps, choices and variables in Sect 2.3 and 2.4 are poorly explained and could be expanded upon. More details can be found in the specific comments.

2.2.4) Reference situation selection (current 2.5.2 – 2.5.4)

2.3) Implementation considerations. It is unclear how VALHALLA can be embedded in a climate model. VALHALLA has a list of insolation situations and a list of snow states, but how do you connect the VALHALLA snow states to the actual snow state in a model? Concerning the incoming radiation, one could simply feed broadband downwelling radiation to VALHALLA, but that is ignoring the spectral radiation available from radiation modules within atmospheric modules? Why should you do that? Furthermore, what input parameters are required, which parameters are then calculated using what equations, are there lookup tables involved, what input parameters are required, what do you do for cases with a SZA larger than 80 degrees, etc. This would be in my opinion a vital subsection and should take away the confusion.

AC: Thanks a lot for this suggestion. The entire method section has been rewritten and restructured according to your comments. We, however, believed that the proposed section 2.3 would fit better in the discussion section. For this purpose, we added a section implementation considerations in the discussion section. The proposed modifications are reported below.

Modification in the methods :

[revised manuscript text omitted]

2. The comparison with Van Dalum et al. (2019), mentioned in the introduction and discussion, is inadequate. The authors have to realize that SNOWBAL and VALHALLA fulfil a different niche. SNOWBAL is a coupling scheme, which is built to couple a spectral albedo model like TARTES with a narrowband radiation scheme as is available in RACMO2. This does not only allow for the physics of TARTES to be directly implemented in a climate model, but also allows for an absorption profile on every timestep for each (sub)surface snow layer and for each narrowband, hence allowing internal heating. VALHALLA, on the other hand, as far as I could tell, does not directly couple TARTES, but is an albedo parameterization, as it approximates the total amount of energy absorbed in the snowpack. It reduces the computational time and applies the physics of TARTES indirectly into a climate model, but loses the information of internal absorption of energy. Hence, it is comparing apples with oranges.

AC: Thanks for your thoughts on this point. We are sorry that the first version of the manuscript does not perfectly reflect the ability of SNOWBAL. The introduction part on SNOWBAL was rewritten according to your comments (see proposed modifications in the response to your comment above).

Comparing the number of *tps* used with the number of RWs is also not a valid comparison, as RWs are not used to determine a kernel function. Also, the number of RWs that are used in SNOWBAL are determined by the number of narrowbands available in the climate model to couple. The uncertainty of SNOWBAL would lower if more narrowbands are available, as the sub-band variability is then reduced. At the contrary, the computational costs of SNOWBAL-TARTES would increase, while it remains the same for VALHALLA. In conclusion, I would suggest to revise or (partly) remove the comparison with SNOWBAL.

AC: The comparison with SNOWBAL and the difference between SNOWBAL and VAHLALLA has been fully rewritten in two sections in the discussion (comparison with existing methods and implementation consideration). The two proposed sections are reported in the response to your general comments 1/ and 3/. Please see also response to Joseph Cook comments.

3. In my opinion, the manuscript would benefit if some analysis is done by comparing VALHALLA with existing albedo parameterizations (like the parameterization of Gardner & Sharp (2010)). It would allow the authors to illustrate the importance of VALHALLA with respect to previous work and validate its necessity. This could also be used to further highlight what processes are now captured properly and could answer some questions. For example, how well does VALHALLA perform on real cases, or alternatively, on snow layers as produced by climate models. Such models typically have many thin but distinctive layers, so how well does it perform then? How would VALHALLA treat ice lenses close to the surface? Adding a short comparison would improve the manuscript.

AC : Thanks for the suggestions. We performed the comparison with Gardner & Sharp parameterization even though the later parameterization and VALHALLA also fulfill different niches since VALHALLA relies on exact multilayer radiative transfer calculations for snow (TARTES in our study). We first compared

the results of the Gardner & Sharp parameterization to SBDART-TARTES calculation at 1 nm spectral resolution. The figure below shows the results obtained for all our atmospheric and snow profiles.

[Figure]

*Figure 1 - Mean calculated broadband albedo errors between Gardner & Sharp, 2010 and SBDART-TARTES albedo computed from 1 nm spectral resolution calculation. The broadband albedo error is represented versus the sun zenith angle in degree.*

The Gardner & Sharp parameterization depicts a small positive bias consistent with figure 11 in Van Dallum et al., 2019. To avoid unnecessary complexity, we decided to report the standard error given in the text of Gardner & Sharp, 2010 (0.01) in Figure 6. The new figure 6 is now :

[Figure]

We also detailed the comparison with Gardner & Sharp in the discussion section (section 4.3 Comparison to other existing methods, see proposed text in the response to general comment 1).

Regarding the comments on the snow layering, the accuracy of VALHALLA depends on the accuracy of the multilayer snow RT model used for the tie points (TARTES here). For ice lenses, since the spectral variations are still highly dependent on the refractive index of ice, when the RT model is able to do a correct calculation at the tie points, then we believe that the accuracy of VALHALLA won't change.

These points are now detailed in the new sections in the discussion (section 4.3, see above and section 4.4 Implementation considerations - see the proposed text in our response to your general comment 1/).

4.  Can the authors say something about or show why they do not deem it necessary to make a distinction between liquid and ice water clouds and why they keep the water vapour content constant?

AC: We guess that the comments on water vapour content being constant is for the reference profiles (Table 2). The choice to keep the water vapour content constant was guided by the fact that method (Eq. 9) accounts for the effect of the water vapour in the atmosphere (not accurately since we used the refractive index of ice and not the one of water vapour, see eq. 6). Separate reference profiles could also be calculated for liquide and ice water clouds, but we did not test it in the study.

The details on the clouds properties have been added in section 2.2. In the discussion we also add a sentence about it : "**The reference profiles can also be adapted to the cloud types (liquid water or ice droplets, droplets radius) when this information is available together with the solar radiation**."

5.  The authors state that VALHALLA could also be used to determine narrowband albedos. However, I am not convinced yet. How well does the method do for small narrowbands, in which, for example, only one or two *tps* are located? Can the authors say more about applying VALHALLA to a narrowband scheme?

AC:  We guess this comment refers to former lines 335-336. We added more information for the results obtained for narrowband inputs in section 4.3 (please see proposed text in response to your general comment 3.). We chose not to detail the results with the narrowband input, since one of the advantages of the method is that it can run for broadband input. This is now explained in sections 4.3 and 4.4 (see proposed text in the response to the general comment 1/).

6.  Some parts of the manuscript would benefit from more interpretation. See specific comments for more details.

AC:  All specific comments below have been accounted for. See below for a point by point answer.

 I also have some specific comments that I would like to see addressed.

**Specific comments**

P1 L16: Please add a reference to this statement.

AC: We add a reference to Warren, 1982 to justify the statement.

P1 L17: Shortly define the albedo here.

AC: The sentences were modified as follows: "**The albedo, defined as the fraction of reflected solar radiation, is very high** for fresh snow **and** limits energy absorption by the snowpack.  Darker or old snow and glacial ice absorb more **solar** energy \citep{warren1982optical, gardner2010review}."

P2 L35-37: "This is usually … and ice surfaces." Climate models are also often limited to narrowbands, so spectral fluxes are not available and consequently the albedo has to be determined for narrowbands or broadband.

AC: Right, the paragraph was changed as follows :

"In addition to the above-mentioned requirements, accuracy in the estimation of the energy absorbed at the snow surface can be achieved through spectral calculation of the albedo but remains numerically expensive. **This also requires spectral calculations of the solar irradiance that are most of the time not available in climate models.**  This is usually overcome in most global and regional climate models by computing broadband or narrowband albedo to estimate the energy budget at the snow and ice surfaces. The broadband albedo is defined as the ratio between total reflected and total incident solar energy integrated across the entire solar spectrum, whereas the narrowband albedo is integrated **over a limited range of the solar spectrum**. These integrations however lead to a bias in the calculation of the snowpack albedo, which ultimately propagates in the computation of the surface energy and mass budgets.``

P2 L38: This might be a bit confusing, as the narrowband albedo is usually defined as the albedo of a spectral band, not over several spectral bands.  AC: This has been changed. Please see the modifications above.

P2 L47: Van Dalum et al. 2019 calculate RWs for the first 12 of the 14 bands of RRTMsw.

P2 L48: It is also important to note that with using the RWs computed with SNOWBAL, Van Dalum et al. 2019 actually couple TARTES with RACMO2. This does not only allow for the calculation of a narrowband albedo, but also provides the absorption of solar radiation in every modeled snow layer for each narrowband.

P2 L53-L56: Although SNOWBAL has been used for RACMO2, it is not true that it is not applicable for other models. RWs can be determined for models with a different set of narrowbands just as easily, as long as the SZA, cloud content and water vapour content are available.

AC: The paragraph was rewritten according to the 3 above comments and now reads :

To overcome these uncertainties while maintaining an adequate calculation time to remain competitive, new methods are developed. One of them, recently developed by (Van Dalum et al., 2019), effectively couples a snow spectral albedo model with a narrowband atmospheric radiation scheme. This method (Spectral-to-NarrOWBand ALbedo module; SNOWBAL) al-

45    lows the coupling of the radiative transfer model TARTES (Two-streAm Radiative TransfEr in Snow, Libois et al., 2013) with the European Centre for Medium-Range Weather Forecasts (ECMWF) radiation McRad scheme based on the shortwave Rapid Radiation Transfer Model (RRTMsw) embedded in RACMO2 (Mlawer et al., 1997; Clough et al., 2005; Morcrette et al., 2008; ECMWF, 2009). They used the first 12 of the 14 predefined representative wavelengths (RWs, for every 14 bands of RRTMsw) dependent on irradiance distribution and albedo within a spectral band to calculate the narrowband albedo and radi-

50    ation absorption in each (sub)surface snow layer. To determine the 12 RWs, a limited number of properties of the atmosphere are selected using a look-up table (LUT). They demonstrate that RWs primarily depend on the SZA, cloud content and water vapour. This method is tested on different types of snow and for clear-sky and cloudy atmospheric conditions, and represents broadband snow albedo with low uncertainties (<0.01). In van Dalum et al. (2020a, b), the SNOWBAL module is evaluated in RACMO2 over the Greenland ice sheet. This method can therefore be used on large surfaces while accurately representing

55    the albedo of snow and ice. The impact of snow properties on the RWs are not accounted for in the LUTs since it is negligible in the case tested in Van Dalum et al. (2019). This might be a limitation when the narrow bands of the atmospheric model are too large. The use of this method with another model than RACMO2 will require a recalculation of the LUTs for a different

P3 L69-73: For me it is not clear why Crocus is introduced here. It is not mentioned in the introduction and as far as I could tell also not used in the remainder of the manuscript. So it might be better to remove this part.

AC: We agree with the reviewer. The part on Crocus was removed and the entire Method section was rewritten according to the general comment 1/ (please see response to general comment for modifications in the manuscript).

P3 L79: Please be aware that SSA is usually the abbreviation of specific surface area. Also add a space between $m^2$ and $kg^{-1}$ and also apply this to all other units in the manuscript. AC: Modified accordingly and also in the rest of the manuscript and in all the figures.

Eq 1: 'SSA' and 'ice' should not be in italics. AC: This has been modified.

All Equations: As an equation is part of a sentence, punctuation is often required at the end of the equation. AC: This has been modified.

P3. L83-84: Please introduce grain shape and 'g' and 'B' a bit more.

AC: The sentences have been changed to : "**We used two shape parameters that are relevant for the optical properties of snow : the asymmetry  parameter $g$ (dimensionless) and the absorption enhancement parameter $B$ (dimensionless, \citealp{libois2013}). $g$ quantifies the amount of radiation that is scattered forward for a snow grain and $B$ quantifies the lengthening of photon paths inside a snow grain due to internal multiple reflections.**"

P4, Sect 2.2: SBDART should be described in more detail in my opinion. Also, why did you specifically use this version of DISORT?

AC: We believe SBDART is a useful tool to simulate spectral solar irradiance. It enables the use of a large number of parameters and has proven to be quite accurate in snow covered areas. The calculation is also not too time demanding even though faster alternatives can be found. The section 2.2 was updated as follows :

"**This atmospheric radiative transfer model was chosen since it provides accurate simulations of solar irradiance in snow covered areas (e.g. \citealp{tuzet2020}) and offers a large number of parameters to set for the atmospheric properties.**"

P4 Sect 2.3: A few introductory sentences are necessary in my opinion to illustrate that you will introduce the physical concepts of VALHALLA.

AC:  We agree that this was missing. A full paragraph was added in the beginning of the new method section (please see also response to your general comments 1/).

This reads :

75 **2 Method**

The VALHALLA method relies on the accurate calculations of the solar radiation absorbed by the snowpack for a small number of selected wavelengths, named tie points ($tps$) in the following. The number of tie points is kept as small as possible to limit the computing resources. Between these tie points, the VALHALLA method interpolates the absorbed radiation based on kernel functions that reflect the main spectral variations of the absorbed radiation across the solar spectrum. The general reasoning of 80 the method consists in assuming that the spectral variation between tie points can be approximated using the refractive index of ice. The calculation of the absorbed radiation at the tie points can be performed with any radiative transfer model. In the following, we selected the Two-streAm Radiative TransfEr in Snow (TARTES, Libois et al., 2013) for the snowpack and the Santa Barbara DISORT Atmospheric Radiative Transfer (SBDART, Ricchiazzi et al., 1998a), for the atmosphere.

**2.1 Radiative transfer models**

P4 L97: The use of the letter 'r' for the spectral albedo is unconventional and may cause confusion, as it is often used for grain radius.

AC: We believe 'r' is conventional for reflectance in optics. We thus did not change the letter, but added the word reflectance in the first sentence.

P4 L97: "… or an homogeneous, …" --> "… or a homogeneous, …" AC: modified

P4 L101-104: On L104 you define Lambda-tilde as $\lambda/\lambda0$. This should be moved to L101. Also, where do you use the absorption coefficient of Eq. 3 and explain the Angstrom coefficient. Furthermore, what does 'pol' in Eq. 3 mean, pollution?

P4 L101: Can you describe the parameter 'l' in more detail? It is a bit vague, as it apparently represents both the size and shape of a grain.

AC: Line 101-107 were fully modified and simplified according to the two comments above. It now reads:

The spectral direct albedo, also called directional-hemispherical reflectance, $r$ of a homogeneous, optically infinite snowpack can be approximated by the following relationship (Libois et al., 2013; Dumont et al., 2017; Kokhanovsky et al., 2018) :

$$r(\lambda) = \exp\left( - u(\mu_0)\sqrt{\frac{64\pi}{3\rho_{ice}\text{SSA}(1-g)}\left(\frac{2n(\lambda)B}{\lambda} + 3\frac{\rho_{ice}}{\rho_{LAP}}c_{LAP}C_{abs}^{LAP}(\lambda)\right)} \right), \tag{2}$$

where $u(\mu_0) = \frac{3}{7}(1+2\mu_0)$; $\mu_0$ is the cosine of the solar zenith angle, $n(\lambda)$ is the imagery part of the ice refractive index at the wavelength $\lambda$, $c_{LAP}$ is the light absorbing particles concentration, $\rho_{LAP}$ is the volumetric mass of the light absorbing particles, and $C_{abs}^{LAP}(\lambda)$ is the absorption cross-section of LAPs.

When neglecting the spectral variations due to the presence of LAPs in Eq. 2, the first order of spectral albedo variations can be approximated as

$$r(\lambda) \sim exp(-J\sqrt{\frac{n(\lambda)}{\lambda}}), \tag{3}$$

where $J = u(\mu_0)\sqrt{\frac{128\pi}{3\rho_{ice}\text{SSA}(1-g)}}B$. $J$ is constant with $\lambda$ and depends only on illumination and snow physical properties.

P4 L104 and elsewhere: Units should not be in italics. AC: Modified everywhere in the paper and in all figures.

P4 L105: "… can be approximate as" --> "… can be approximated as" AC: Modified

Eq 4: Why can Eq 2 be approximated as Eq 4? Please explain. AC: we gave more details (please see modifications in the response to your comment on line 101).

P4 L107: Please use one symbol for the imaginary part of the ice refractive index, as a different symbol is used at L100. AC: Thanks for noticing it, this was modified, now only n is used everywhere.

P4 L107 and Eq 4: Is the parameter 'J' a tuning parameter of some sort? Please explain in more detail.

AC: This was modified and explained. The value of J according to equation 2 was also added in the text. See proposed modifications in response to your comment on page L101.

P4 L109: "The fraction of absorbed energy in the snowpack …" --> "The fraction of absorbed energy with respect to the incoming energy…", or something similar. AC: Changed

P4 L112: "… through a given media…" --> "… through a given medium" . AC: Changed

Eq. 7: Similar to Eq 4, what did you assume to rewrite Eq. 6 to Eq. 7? Also similar to the parameter 'J' of Eq 4, what is parameter 'D' here?

AC: The description of D was wrong, sorry. This was changed to :

$$I(\lambda) = I_0(\lambda) \exp(-c_a(\lambda)L), \qquad (5)$$

$c_a(\lambda)$ is varying with the atmospheric profile, i.e. with the aerosols properties, the properties of gas in the atmosphere (water vapour, ozone, ...), the solar zenith angle and the cloud properties. Here we assume that the spectral variations of the solar irradiance at the surface can be written:

$$\quad I(\lambda) \sim E_{ref}(\lambda) \exp\left(-D\sqrt{\frac{n(\lambda)}{\lambda}}\right), \qquad (6)$$

where $E_{ref}$ is the total incident energy at the wavelength $\lambda$ for a reference atmospheric profile and $D$ is constant with wavelength. In other words, this means that for instance, for a given location, we assume that changes in the atmospheric solar irradiance with time can be modelled using Eq. 6, i.e. main spectral changes are driven by the water vapour (assuming that the refractive index of water vapour is close to $n(\lambda)$).

P5 L125 – 126: "(varying between 10 and 80)", missing units. AC: units added

P5 L129: What do you mean with 'experience' irradiance? Please see the answer to the next comment.

Eq. 9: It is not clear over what you are integrating, I suppose wavelength? If so, add 'dy'. Also, why do you want to calculate 'C', what does it represent? Furthermore, what is 'i', please specify.

AC: We agree that this was confusing, this has been clarified as follows :

The method uses a reference irradiance profile with a spectral resolution of 1 nm, $E_{ref}$. For each SZA value (varying between 10 and 80 degrees), a reference irradiance profile is calculated with SBDART. In total, 16 reference profiles were used (one set for clear-sky and partially cloudy conditions and the other one for full overcast conditions; see Section 2.5.2). These

135 profiles are used to calculate a coefficient $C$ between the broadband reference irradiance $E_{ref}$ (integral of the reference profile) and narrow or broadband irradiance irradiance $E_{exp,i}$ given by an atmospheric model for each narrow or broad spectral band $i$:

$$C_{b_i}^{b_{i+1}} = \frac{E_{exp,i}}{\int_{b_i}^{b_{i+1}} E_{ref}(\lambda)d\lambda}, \qquad (8)$$

where $b_i$ and $b_{i+1}$ are the max and min wavelengths of the bands in which the atmospheric model is providing the solar

140 incident radiation. $C_{b_i}^{b_{i+1}}$ thus represents the scaling factor between the incident radiation provide by the atmospheric model and the reference irradiance for each narrow or broad spectral band of the atmospheric model. In the following, we use $b_i = 320$ nm and $b_{i+1} = 4000$ nm. Thus we assume that the broadband incident radiation only is available from the atmospheric model.

Eq 13: I think there is an error in this equation. The 'G' term should be in the denominator. I quickly checked the VALHALLA code and it seems that it is correct there, but please verify this.

AC: Thanks for spotting this error, yes 'G' should be in the denominator. This has been corrected in Eq. 13.

P6 L147: This is a strange sentence, please rewrite and also explain in more detail what happens in Eq. 14. Do you do this optimization for all cases? Please also define the Delta term.

AC: The sentence was indeed strange, it has been rewritten as follows adding details about delta and Eq. 14 : "**Namely, an optimization method is used to solve Eq. \ref{eq:method}. The optimization algorithm is finding the value of $G_{tp_n}^{tp_{n+1}}$ for which $\Delta_{tp_n}^{tp_{n+1}}$ is the closest to zero, $\Delta_{tp_n}^{tp_{n+1}}$ being the difference between the left and the right sides of Eq. \ref{eq:method}.**"

Sect. 2.5.1: It might be beneficial if this is told earlier, i.e. in Sect. 2.4.

AC: This information was moved earlier at the beginning of the method section (please see response to general comment for the proposed modifications).

P6 L150-151: "30 tps is selected" à "30 tps are selected". AC: Changed

Fig 1: Can you explain why there is no tp at the major minima of the spectrum, like around 1200 and 1400 nm? Furthermore, the units should not be in italics.

AC: the units have been changed in all the figures. The major minima were not necessary (after test) to improve the accuracy of the method.

Sec. 2.5.2 and 2.5.3: These sections should be merged. Could you also specify for cloudy conditions if you use a liquid water or ice cloud and what its droplet radius is.

AC: In response to your general comment 1/, the two sections have been merged in a new section 2.2.3 Numerical settings. See response to the general comments 1/ for the modifications of the whole section methods.

We also add the information on the type of cloud and droplet radius in the new section, it reads:

"The main SBDART input parameters used in this study are the aerosol optical depth (AOD),  cloud-layer optical depth ($\tau$), boundary-layer aerosol type selector (IAER) and SZA (Table \ref{table:2}). **For the cloud properties, we used liquid water droplets with a radius of 8 microns.**"

P7 L161-163: "These parameters … on model outputs". I do not fully understand this.

AC: The two sentences were changed to : "**These parameters have been selected after a principal component analysis of the spectral absorbed energy. The principal component analysis aimed at obtaining  a list of representative parameters with the most pronounced influence on the absorbed energy spectrum.**"

Table 2: "one fore full-overcast" à "one for full-overcast". I also suppose that the Mid-latitude winter atmospheric profile is one of the AFGL standards, please specify.

AC: The caption of Table 2 was corrected and we added the information about the AFGL standards.

Table 3: Not all units are correct in this table and the last sentence of the header should be reformulated. AC: All the units have been corrected and the last sentence of the header was reformulated to: "**For layer 3, the values of all input parameters, besides soot and dust contents, are constant. For layer 4, all input parameters are constant.**"

Sect. 2.5.4: This section is hard to follow and some information is missing; it would be beneficial if it is rewritten. Furthermore, can you say why you consider these impurity concentrations? They seem very high to me. Similarly, the SSA of 155 $m^2$ $kg^{-1}$ for fresh snow looks very high.

Also, what grain shape did you assume (so what g and B in TARTES) and which refractive index data set did you use (e.g., Warren and Brandt (2008))? Please elaborate on this.

AC: Yes 155 for new snow is very high but has been measured (e.g. Domine et al., 2007). The sentence was rephrased since it was confusing. All the other informations and justifications about the range and values selected has now been added in the section that reads :

**TARTES settings**

195    The main TARTES input parameters used in this study are the surface specific area (SSA) of the first layer of the snowpack, the snow water equivalent (SWE) for each layer of the snowpack and the LAPs concentration. We consider a snowpack with 3 layers of varying thickness and density (Table 3). These layers represent at most the first 20 centimeters of the snowpack, whose physical properties largely determine the albedo of the snow. The principal parameters on albedo calculation are the SSA of the first layer, the SWE of the first layers and the LAPs concentration of two first layers of the snowpack. We selected

200    a wide range of SSA values (2 to 155 $m^2kg^{-1}$) in order to cover most of the snow types found on Earth (Domine et al., 2007). SWE gives the mass of snow and is the product between thickness ($t$) and density ($d$). For pure snow (without LAPs), the SWE values are provided for the first three layers of the snowpack. For snow with LAPs, the SWE and LAPs concentration (for soot and dust) are provided for the first two layers of the snowpack. For layer 3 and layer 4, the values of all input parameters are fixed. The ranges of LAPs content for soot and dust have been selected to cover the wide range of conditions that can be

205    encountered in the various regions of the world from almost pristine snow in Antarctica to highly polluted snow (see e.g. Table 2 in Tuzet et al. (2020) for soot and Sterle et al. (2013) for dust). As recommended by Libois et al. (2014), we set the shape parameters $B$ and $g$ to 1.6 and 0.85. The refractive index from Warren and Brandt (2008) was used.

Fig 2: The wrong section numbers are shown here: "… in Section 5.c) and 5.d) using …" Changed

P8 L185-186. "Overall, the median error on the broadband absorbed energy calculated for all simulations decreases with increasing SZA". Can you explain why?

AC: This is because the incoming (and thus the absorbed) energy is decreasing with increasing SZA. And p 8 L185-186 has been modified as follows : "Overall, **the broadband albedo biases vary little with SZA and the biases of the absorbed energy decrease with SZA. This is consistent with higher absorbed energy for lower SZA (higher incoming radiation and lower albedo).**"

P11 L203: There is a redundant dot here: "Figure 3a,b. show the…" AC: Modified

Fig 3: Not all units are correct and change "for all the simulations described 2.5.3 and 2.5.4 …" into "for all the simulations described in Sect. 2.5.3 and 2.5.4…".  Please define 'BB error'.

AC: Units in Figures 3, 4 and 5 have been corrected. The reference to the section has been corrected and BB error defined.

P11 L207: "More than 75% of the errors are positive". Please add one time what you mean with 'positive' or 'negative' errors, to take away any possible confusion, i.e., that VALHALLA overestimates the absorption of energy for a positive error.

AC: Thanks, this has been modified to :"More than 75$\%$ of the errors are positive, **meaning that VALHALLA overestimates the absorbed energy.**"

P11 L217: "'the majority of the errors are positive and…", I think 'positive' should be 'negative' here.

AC : Yes, modified.

P14 L250: What do you mean with "the most important errors"?

AC: The sentence has been modified to : "LAPs being highly absorbent at the beginning of the spectrum (between 0.3 and 0.8 $\si{\micro\meter}$, \citealp{warren1982optical}), **the highest spectral absolute errors** are consequently located in this wavelength range."

P14 L258: Typo AC: Modified

Fig 6: Can you be a bit more specific about what you mean with 'Method' in the right panel? Can you also state the meaning of the boxes, whiskers etc. or refer to a previous figure where you explained it.

AC: Method has been changed to VALHALLA in Figure 6 (see above for the new version of the figure). The caption has been changed :

**Figure 6.** Errors on the broadband albedo for different constant spectral resolutions (left) and comparison with the errors of our method (right). For these resolutions, the broadband albedo is computed by TARTES-SBDART and is compared to the one computed at 1 nm resolution. The red lines indicate the median (same as in Fig 2), the box shows the 25th to 75th percentiles and the whiskers show the 5th to 95th percentiles. The grey dotted lines correspond to an error of $\pm 0.01$ as in Gardner and Sharp (2010).

Sect. 3.5: This part is hard to follow. For example, what do you mean with "the broadband albedo is calculated between 320 and 4000 nm and then between 327 and 4000 nm"? Furthermore, can you interpret Fig. 6 a bit more. For example, why does a higher resolution generally result in a negative bias? Why is the bias at a 72 nm resolution so much more negative than the bias of a bit lower and higher resolution? Please rewrite this section.

AC: The comparison between 320 and 327 nm was removed for more clarity and section 3.5 was fully rewritten to :

"In Figure \ref{fig:6} we compared the broadband albedo bias obtained with the VALHALLA methods to the bias obtained for varying constant spectral resolution. The comparison was performed using the simulations from section \ref{subsec:numerical}. For constant spectral resolution, the absolute bias generally increases with the spectral steps and tends to be more negative. This means that the bias on the absorbed energy tends to be more positive when the spectral steps increase. We believe that for large spectral resolution, the integration over the spectrum is missing the absorption bands leading the integral to be higher than for smaller spectral steps (see e.g. the spectrum in Figs. \ref{fig:3}-\ref{fig:5}).** The VALHALLA method presents biases on the broadband albedo with absolute difference lower than 0.005 which are comparable to the bias obtained with resolutions lower or equal than 20 $\si{\nano\meter}$ (reference resolution used at MétéoFrance in research activities). The method uses 30 $tps$ against 184 wavelengths for a calculation at 20 $\si{\nano\meter}$ resolution. However, for the same bias on the broadband albedo, the method thus uses six times fewer bands than a calculation at 20 $\si{\nano\meter}$ resolution."

We hope this is clearer this way.

P16 L278-L280: "The irradiance provided by the method must, therefore, be as close as possible to the irradiance of the exact calculation to obtain a good representation of the absorbed energy.". What does this mean if you apply your method into a climate model, as the irradiance is often not as close to the exact calculation in such model.

AC: The sentence was confusing and misleading and was consequently removed.

P17 L288-290: It is maybe also good to mention that although the absolute error decreases with SZA, the relative error generally increases for high SZA, as can be seen in Figures 3-5. Furthermore, for very high SZA (>85 degrees), what do you use then?

AC: yes, we agree. This was reformulated as follows :

"The associated absolute error evolves as a function of the amount of energy absorbed by the snowpack and is therefore driven by the absorbing factors such as the SZA and the SSA. The error on the absorbed energy, therefore, increases with a decrease in the solar angle and a decrease in the SSA value of the first layer of the snowpack. **Although the absolute error decreases with SZA, the relative error generally increases for high SZA, as can be seen in Figures \ref{fig:3}-\ref{fig:5}. For SZA higher than 85$^\circ$ (not tested here), the broadband albedo might be interpolated between the value at 85$^\circ$ and 1. The choice of an adequate reference irradiance profile for the simulation globally determines the accuracy of the absorbed energy error calculated by VALHALLA**. However, the choice of $tps$ is also a determining factor in a good estimate of the energy absorbed by the method."

P17 L308-310: "The method presented … in van Dalum et al. (2019)". SNOWBAL is not only adapted to RACMO2, but can be applied to any narrowband climate model. Furthermore, SNOWBAL is also based on spectral albedo calculations. For P17 L305-314: As is stated in the general comments, comparing SNOWBAL and VALHALLA is like comparing apples with oranges in my opinion. AC:  lines 305 to 314 have been fully rewritten according to your comments, please see the proposed text in response to general comment 3.

P18 L329-330: "For the other … profile is inadequate." This sentence is confusing, please reformulate. AC: The sentence was reformulated to : "**The use of reference profiles with an adequate value of SZA is necessary to the good accuracy of the method.**"

---

## Referee Report (RR1)

I am glad to see that the authors have addressed most of the previously raised issues very thoroughly and the manuscript has improved considerably. There are still a few rather minor issues, most of them textual suggestions. After these issues are solved, the manuscript is ready for final publication in my opinion.

1. In the previous review, I wrote a few comments about the comparison of VALHALLA with SNOWBAL. Most issues are sufficiently resolved, but one important topic still remains. An important difference between SNOWBAL and VALHALLA, from my understanding, is that when implemented in a climate model, SNOWBAL is also able to provide solar radiation absorption for each subsurface snow layer on each model time step, thus allowing for internal heating. VALHALLA provides the total energy absorption of the snowpack with high accuracy, but if I am not mistaken, does not provide shortwave radiation absorption for each subsurface snow layer. Hence, a climate model is then unable to determine internal heating.
In other words, the answer of the following question should help take away the confusion: 'When implemented in a climate model, does VALHALLA provide solar radiation absorption for every subsurface snow layer and on every time step?'.
If the answer is yes, I think this should be mentioned in the manuscript. If the answer is no, then this should be mentioned in the comparison with SNOWBAL where the purpose of SNOWBAL is compared with VALHALLA.

2. P2 L35-38: Some citations would be useful here.
3. P5 L124: '… only on illumination …' → '… only on SZA …'
4. P6 L153: '… or broadband irradiance irradiance …' → '… or broadband irradiance …'
5. In Sect. 3.5, it is not immediately clear what is meant with 'constant spectral resolution', especially in P15 L293-294: '… obtained for varying constant spectral resolution.'.
6. P15 L296: 'We believe that for large spectral resolution…'. This should be low resolution.
7. P18 L334: 'The tps were taken into account in the method (30 tps), …'. I don't understand this.
8. P19 L366: 'ration' → 'radiation'

The following sentences can be reformulated to improve readability:
1. P15 L289-290: '… is absorbing a large amount of energy, such as with an important LAPs concentration, the biases on…'.
2. P15 L301-302: 'However, for the … at 20 nm resolution.'

---

## Author Response (AR2)

**Authors response to review of gmd-2020-442 by C.T. van Dalum**

Reviewer comments are in black. Author response in blue and proposed changes in the manuscript in **bold blue** or in latex fonts.

I am glad to see that the authors have addressed most of the previously raised issues very thoroughly and the manuscript has improved considerably. There are still a few rather minor issues, most of them textual suggestions. After these issues are solved, the manuscript is ready for final publication in my opinion.

The authors are very thankful to Christiaan van Dalum for re-reviewing the manuscript and providing helpful comments. All the comments have been taken into account and a point-by-point response is provided below.

1. In the previous review, I wrote a few comments about the comparison of VALHALLA with SNOWBAL. Most issues are sufficiently resolved, but one important topic still remains. An important difference between SNOWBAL and VALHALLA, from my understanding, is that when implemented in a climate model, SNOWBAL is also able to provide solar radiation absorption for each subsurface snow layer on each model time step, thus allowing for internal heating. VALHALLA provides the total energy absorption of the snowpack with high accuracy, but if I am not mistaken, does not provide shortwave radiation absorption for each subsurface snow layer. Hence, a climate model is then unable to determine internal heating. In other words, the answer of the following question should help take away the confusion: 'When implemented in a climate model, does VALHALLA provide solar radiation absorption for every subsurface snow layer and on every time step?'. If the answer is yes, I think this should be mentioned in the manuscript. If the answer is no, then this should be mentioned in the comparison with SNOWBAL where the purpose of SNOWBAL is compared with VALHALLA.

In its current implementation, described in the paper, VAHLALLA doesn't directly provide access to the broadband absorbed energy in each layer, but it does provide the absorbed energy in each layer for each tie point. The step of the interpolation of the absorbed energy per snow layer has not been performed yet, but it should not be too complicated (analytical formulas from Kokhanavosky et al., can also be used for that). To make this point clearer we performed the following modifications in the manuscript:

Section 4.3 was modified as follows :

"The SNOWBAL coupling scheme from Van Dalum et al. (2019) described in the introduction of our study provides albedo calculation with an accuracy better than 0.01. Thanks to the physics of the snow radiative transfer model TARTES, SNOWBAL accurately calculates the vertical distribution of the absorbed energy in the different snow layers. **For** VALHALLA when using the method with TARTES or with any other multilayer radiative transfer model for snow (e.g. SNICAR, He et al., 2018)**, the vertical distribution of the absorbed energy is calculated for the tps but the vertical profile of broadband absorbed energy is not directly available. This would require further development of the method.**"

2. P2 L35-38: Some citations would be useful here.

References to Gardner & Sharp, 2010; Munneke et al., 2011 were added :

"This is usually overcome in most global and regional climate models by computing broadband or narrowband albedo to estimate the energy budget at the snow and ice surfaces **(Gardner & Sharp, 2010 ; Munneke et al., 2011)."**

3. P5 L124: '… only on illumination …' à '… only on SZA …' Changed

4. P6 L153: '… or broadband irradiance irradiance …' à '… or broadband irradiance …'Changed

5. In Sect. 3.5, it is not immediately clear what is meant with 'constant spectral resolution', especially in P15 L293-294: '… obtained for varying constant spectral resolution.'.

Yes, we agree that it was not clear. This was changed as follows :

"\subsection{Comparison to calculations **with regular spectral resolution**}

In Fig. \ref{fig:6} we compared the broadband albedo bias obtained with the VALHALLA methods to the bias obtained for **spectral resolution varying from 2 to 100 nm**. The comparison was performed using the simulations from section \ref{subsec:numerical}. For **regular spectral resolution**, the absolute bias generally increases with the spectral steps and tends to be more negative. This means that the bias on the absorbed energy tends to be more positive when the spectral steps increase. We believe that for **low** spectral resolution, "

6. P15 L296: 'We believe that for large spectral resolution…'. This should be low resolution. Changed

7. P18 L334: 'The tps were taken into account in the method (30 tps), …'. I don't understand this.

The sentence was removed since it was confusing and the beginning of the paragraph was changed to:

"**The accuracy of the method is sensitive to the locations and to the number of $tps$.** "

8. P19 L366: 'ration' à 'radiation'Changed

The following sentences can be reformulated to improve readability:

1. P15 L289-290: '… is absorbing a large amount of energy, such as with an important LAPs concentration, the biases on…'.

The sentence was changed to : "**The biases on the spectral and broadband energy increase with the amount of energy absorbed by the snowpack**".

2. P15 L301-302: 'However, for the … at 20 nm resolution.'

"The **VALHALLA** method uses **30 wavelengths (30 tps) when the calculation at 20 nm resolution requires 184 wavelengths**. **Thus**, for the same bias on the broadband albedo, the **VALHALLA** method uses six times fewer bands than a calculation at 20 $\si{\nano\meter}$ resolution."